# Post-translational modification patterns on β-myosin heavy chain are altered in ischemic and nonischemic human hearts

Maicon Landim-Vieira[1†], Matthew C Childers[2†], Amanda L Wacker[3], Michelle Rodriquez Garcia[1], Huan He[1,4], Rakesh Singh[1,4], Elizabeth A Brundage[5], Jamie R Johnston[1], Bryan A Whitson[6], P Bryant Chase[7], Paul ML Janssen[5], Michael Regnier[2], Brandon J Biesiadecki[5], J Renato Pinto[1], Michelle S Parvatiyar[3*]

[1]Department of Biomedical Sciences, College of Medicine, The Florida State University, Tallahassee, United States; [2]Department of Bioengineering, College of Medicine, University of Washington, Seattle, United States; [3]Department of Nutrition and Integrative Physiology, The Florida State University, Tallahassee, United States; [4]Translational Science Laboratory, College of Medicine, The Florida State University, Tallahassee, United States; [5]Department of Physiology and Cell Biology, College of Medicine, The Ohio State University, Columbus, United States; [6]Department of Surgery, College of Medicine, The Ohio State University, Columbus, United States; [7]Department of Biological Science, The Florida State University, Tallahassee, United States

*For correspondence:
mparvatiyar@fsu.edu

†These authors contributed equally to this work

Competing interest: The authors declare that no competing interests exist.

**Abstract** Phosphorylation and acetylation of sarcomeric proteins are important for fine-tuning myocardial contractility. Here, we used bottom-up proteomics and label-free quantification to identify novel post-translational modifications (PTMs) on β-myosin heavy chain (β-MHC) in normal and failing human heart tissues. We report six acetylated lysines and two phosphorylated residues: K34-Ac, K58-Ac, S210-P, K213-Ac, T215-P, K429-Ac, K951-Ac, and K1195-Ac. K951-Ac was significantly reduced in both ischemic and nonischemic failing hearts compared to nondiseased hearts. Molecular dynamics (MD) simulations show that K951-Ac may impact stability of thick filament tail interactions and ultimately myosin head positioning. K58-Ac altered the solvent-exposed SH3 domain surface – known for protein–protein interactions – but did not appreciably change motor domain conformation or dynamics under conditions studied. Together, K213-Ac/T215-P altered loop 1's structure and dynamics – known to regulate ADP-release, ATPase activity, and sliding velocity. Our study suggests that β-MHC acetylation levels may be influenced more by the PTM location than the type of heart disease since less protected acetylation sites are reduced in both heart failure groups. Additionally, these PTMs have potential to modulate interactions between β-MHC and other regulatory sarcomeric proteins, ADP-release rate of myosin, flexibility of the S2 region, and cardiac myofilament contractility in normal and failing hearts.

## Editor's evaluation

This article surveys differences in the heavy chain of the contractile protein β-myosin in normal hearts and hearts in cardiac failure. This is important in view of its possible regulatory roles in generating contraction. The findings are then substantiated by functional simulations of the contractile process.

## Introduction

The sarcomere is the smallest functional unit in striated muscle. The cardiac sarcomere is composed of thick and thin filament proteins that work together to generate force and shorten the sarcomere, and regulate sarcomere contraction and relaxation in a $Ca^{2+}$-dependent manner (*Gordon et al., 2000*). The cardiac thick filament is composed of myosin II polymers accompanied by associated proteins, myosin-binding protein C (MyBP-C), titin, and obscurin (*Gordon et al., 2000*; *Craig and Woodhead, 2006*; *Wang et al., 2018*). The major proteins of the cardiac thin filament include F-actin-containing myosin-binding sites – tropomyosin (Tm) and the cardiac troponin complex (cTn) (*Gordon et al., 2000*). Myosin molecules of the thick filaments are constituted of six noncovalently associated polypeptides: two heavy chains and four light chains (LCs). The C-terminus of the two myosin heavy chains (MHCs) forms an α-helical coiled-coil tail that extends toward the center of the thick filament backbone. The paired, N-terminal heads of the two MHCs are positioned at the surface, facilitating interactions with actin filaments. Between the heads and the tail is a coiled-coil that makes up the myosin S2 rod segment. The neck region of myosin is the site of accessory protein binding that consists of two pairs of LCs: essential (ELC) and regulatory (RLC). Near the N-terminus, the individual MHC forms a distinct globular structure – myosin S1 fragment – that interacts with the actin filament in a cyclic fashion (*Geeves and Holmes, 1999*). Myosin binds to actin and ATP and undergoes several conformational changes that are essential to its function: (1) actomyosin complex formation along with release of $P_i$ and ADP from myosin cause a swinging motion during the force-generating powerstroke; (2) during rigor, ATP binding to myosin results in a reduction in actin affinity and dissociation of actomyosin; and (3) ATP hydrolysis occurs when myosin is dissociated from actin and primes the lever arm for the next cross-bridge cycle (*Tang et al., 2016*).

Post-translational modification (PTM) of myofilament proteins can regulate their mechanical properties and modulate cardiac sarcomere function. PTMs have been shown to alter the canonical structure, function, localization, and half-life of modified sarcomeric proteins (*Mnatsakanyan et al., 2018*). These modifications can instigate downstream effects on the functional properties of the myocardium, thus providing a rapid, efficient, and energetically favorable mechanism to alter contractile function compared to isoform switching. The context dependence of a modification may also be important as it may be influenced by other PTMs on the same protein or other proteins. Furthermore, PTMs on sarcomeric proteins may be inert under normal conditions, but their functional importance can become evident alongside pathological conditions. Therefore, these normally 'silent' PTMs represent novel targets for therapeutic intervention (*Sumandea and Steinberg, 2011*). Localized, spatially confined pools of kinases, acetyltransferases, and other protein modifiers have been identified as essential for the efficient modification of myofilament proteins. Both histone acetyltransferase (HAT; p300/CBP-associated factor [PCAF]) and histone deacetylase 4 (HDAC4) have been found localized in the sarcomeric matrix (*Samant et al., 2015*; *Gupta et al., 2008*). Furthermore, HDAC6 has also been shown to assume a sarcomeric localization (*Demos-Davies et al., 2014*). Of the PTMs identified on sarcomeric proteins, phosphorylation has been the most extensively characterized. However, acetylation, methylation, oxidation, SUMOylation, and ubiquitination have been reported as well, extending the repertoire of potential modifiers of the sarcomere (*Cui et al., 2014*; *Terman and Kashina, 2013*).

A number of sarcomeric proteins are phosphorylated, including MyBP-C, troponin T (TnT), and troponin I (TnI) by cAMP-dependent protein kinase (PKA), and myosin regulatory light chain (RLC) by myosin light-chain kinase (MLCK) (*Sumandea and Steinberg, 2011*; *Colson et al., 2010*; *Seguchi et al., 2007*; *Chan et al., 2008*). Several kinases are implicated in phosphorylation of striated muscle Tm, including tropomyosin kinase, PKA, and protein kinase C ζ (*Heeley et al., 1989*; *Mak et al., 1978*; *Montgomery and Mak, 1984*; *Reddy et al., 1973*; *Wu and Solaro, 2007*). Remarkably absent from the list of phosphorylated sarcomeric proteins in human hearts is β-MHC, encoded by *MYH7*, the predominant isoform of myosin in the adult human heart. The massive size of β-MHC, ~223 kDa, imposes limitations with current technological approaches and challenges our ability to obtain complete sequence coverage. In a study by Kawai et al. examining PTMs on *MYH6*, the predominant murine myosin isoform,(*Zhang et al., 2020*) phosphorylation sites were identified in control hearts with a number of these residues not phosphorylated in HCM hearts (*Kawai et al., 2017*). In addition, Jin et al. identified acetylated, methylated, and trimethylated residues in human β-MHC using size-exclusion chromatography (SEC)/middle-down mass spectrometry (MS) (*Jin et al., 2017*). Overall, detection of low-abundance PTMs on large proteins has remained elusive and protein enrichment

strategies before liquid chromatography-mass spectrometry (LC/MS) analysis or even greater instrument sensitivity can increase signal intensity of modified proteins (*Mnatsakanyan et al., 2018*; *Zhao and Jensen, 2009*).

While many earlier studies were conducted in animals of varying species, examining changes in PTMs on sarcomeric proteins in the context of human heart disease may provide clues toward potential interventions. Identification of PTMs on β-MHC is the first step in discovering whether these modifications provide beneficial or adverse impacts on cardiac function and disease progression. With mass spectrometry techniques increasing in sensitivity, we are becoming more successful at uncovering even low-abundance PTMs that may have a significant impact on cardiac outcomes. As new reports emerge documenting the presence of PTMs in human sarcomeric proteins, the ensuing challenge remains: testing their in vivo functional consequences. Given the emerging importance of understanding the potential role of PTMs on cardiac muscle performance, we used LC/MS and MD simulations to investigate novel PTMs sites in key functional regions of β-MHC in human hearts in healthy and diseased states. This article provides insight into the potential of these modifications to fine-tune cardiac myofilament performance.

## Results

### Human heart data bank

The patient groups from whom explanted heart tissues were collected included explanted hearts from healthy donors (nonfailing (NF)) and end-stage heart failure patients, who were reported to have either ischemic heart failure (I-HF) or nonischemic heart failure (NI-HF). The demographics for the patients included information on age, gender, and race. Patients' ages ranged from 41 to 69 years with each condition represented (NF, NI-HF, and I-HF) (*Supplementary file 1*) and were similarly distributed among the three groups. In an attempt to provide gender and race balance, one female was included in each group and at least one African American (*Figure 1*). We utilized mass spectrometry to investigate whether PTMs could be found on β-MHC isolated from these hearts and utilized MD simulations to better understand the functional impact these PTMs may have on human cardiac disease presentation. For workflow, see the schematic in *Figure 1—figure supplement 1*.

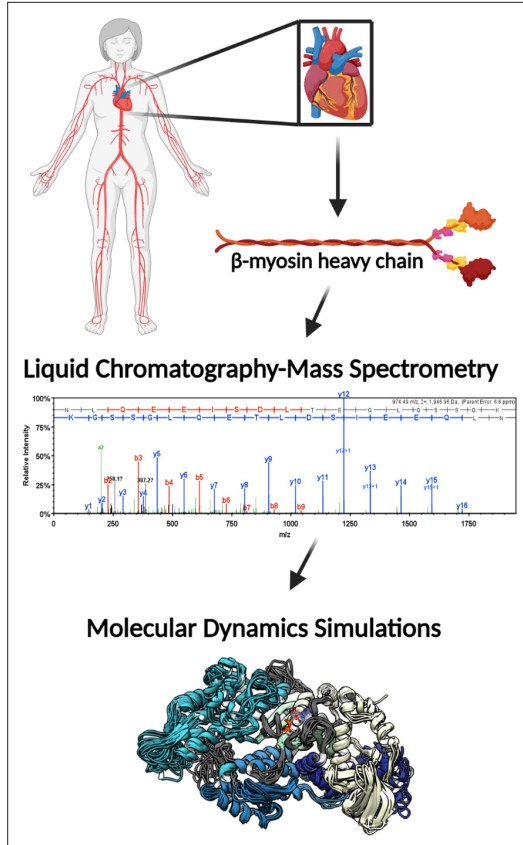

**Figure 1.** Illustrative schematic of the integrative approaches used to identify novel post-translational modifications (PTMs) on human β-myosin heavy chain (β-MHC) and investigate their roles in cardiac muscle regulation. De-identified human heart samples were obtained from nonfailing, ischemic heart failure, and nonischemic heart failure patients. The presence of PTMs on human β-MHC was confirmed by liquid chromatography-mass spectrometry (LC/MS). b-ions and y-ions indicate N-terminal and C-terminal ions, respectively. Molecular dynamics (MD) simulations were used to further understand the functional significance of the newly identified PTMs on β-MHC. The schematic was generated using BioRender.com.

The online version of this article includes the following source data and figure supplement(s) for figure 1:

**Figure supplement 1.** Coomassie-stained gel of homogenized human heart tissues.

**Figure supplement 1—source data 1.** Source data for *Figure 1—figure supplement 1*.

## Identification and location of PTMs on human cardiac β-MHC

Unique high-confidence peptides bearing PTMs were identified using bottom-up mass spectrometry data from NF, ischemic, and nonischemic failing human hearts (*Figures 2 and 3*). The samples were purified using SDS-PAGE and were digested using in-gel trypsinization. We report eight PTMs: six acetylated lysines and two phosphorylated residues, one serine and one threonine: K34-Ac, K58-Ac, S210-P, K213-Ac, T215-P, K429-Ac, K951-Ac, and K1195-Ac (*Tables 1 and 2*). Six of these PTMs are distributed throughout the myosin motor domain (*Figure 4A*) and two within the tail (*Figure 4B*). The myosin motor is comprised of four domains: the N-terminal domain, the upper and lower 50 kDa domains, and the converter arranged around a central β-sheet (*Figure 4A*). The structure of myosin gives rise to three functional regions: the actin-binding cleft, which is formed between the upper and lower 50 kDa domains and interacts with the thin filament; the nucleotide-binding pocket, which is comprised of several regulatory loops that coordinate the nucleotide and transmit structural information throughout the structure; and the converter domain, which converts small-amplitude changes in the motor domain into large-amplitude lever arm motions.

A schematic of the β-MHC protein is included in *Figure 4—figure supplement 1* and indicates key regions of the protein along with the newly identified PTMs. In *Figure 4A*, the three-dimensional structure of the human β-MHC sequence complexed with Mn-AMPPNP (PDB: 4DB1) was used to model the PTM sites at K58-Ac and doubly modified peptide K213-Ac/T215-P. This structure represents a post-rigor ATP-bound state of myosin and was used to assess proximity of the modified residues to structurally important regions in the myosin motor domain. An S2 fragment structure (PDB: 2FXM) was used to model K951-Ac.

Residue K34 is located between the SH3-like domain and the converter domain (*Figure 4A*). K58 is located within the SH3-like domain, which typically serves as a module for protein–protein interactions. Therefore, it is anticipated that acetylation of either K34 or K58 may interfere with normal interactions of this domain. Residues S210, K213, and T215 are sequentially close in the primary structure, and it is interesting to note that they lie together on one face of the myosin motor domain near the ATP-binding pocket in a region called loop 1. This loop is notable for its influence on ATP/ADP cycling (*Murphy and Spudich, 1998*). It is plausible, therefore, to expect that these modifications may influence ATP binding and/or Pi and ADP-release dynamics (*Figure 4A*). S210-P was found to exist as a single modification only. The location and dynamics between modifications of residues K213-Ac and T215-P are interesting as they appear to be co-modifications. Although K213-Ac was only found coincident with phosphorylation of T215, T215-P was also found on its own, without K213-Ac. The K213-Ac/T215-P-modified sequence (amino acids 207–234, *m/z* 766.36) was detectable as an isotope dot product (idotp) with a value below 0.5, and therefore was unquantifiable (*Figure 4—figure supplement 2*). The acetylated residue K429 is located within the myosin head-like domain at the actin-binding interface. The remainder of the modified residues K951 and K1195 are located within the coiled-coil of the S2 region, although K1195 is missing from the model in *Figure 4A*. The panels within *Figure 4B* show closeup views of secondary structural elements and side-chain interactions that neighbor the K58-Ac, K213-Ac, T215-P, and K951-Ac modifications.

## Normalized peak areas of the PTM sites

To investigate the potential significance of these newly identified PTMs for cardiac function, we assessed their abundance in the human heart samples we analyzed. The relative abundance of the modifications was determined by calculating peak areas of modified peptide and normalizing to IRP peak area (amino acids 1504–1521, *m/z* 974.49). The MS/MS spectrum of the trypsin-digested common IRP is shown in *Figure 5—figure supplement 1*, and b-ions indicate N-terminal fragment ions and y-ions indicate C-terminal fragment ions.

Overall, the ratios of modified PTM sites were variable, ranging between 1 and 14 in the NF donor hearts, with a tendency for decreased abundance in failing hearts. In *Figure 5*, the ratio of modified/IRP is shown for peptides 1 (K34-Ac), 2 (K58-Ac), 3 (K429-Ac), 4 (K951-Ac), and 5 (K1195-Ac). Beneath each histogram are the respective tryptic peptides where the reported PTMs were found. The residues shown in brackets are the trypsin digestion sites. Of interest is PTM K951-Ac as acetylation at this site is significantly decreased with ratios of approximately 14 in control hearts and 5 in failing hearts. Following the same trend (but not statistically significant) is K1195 also in the tail region. Both K34-Ac and K429-Ac are found in the myosin motor domain where they may be protected from HDAC activity. K429-Ac is a

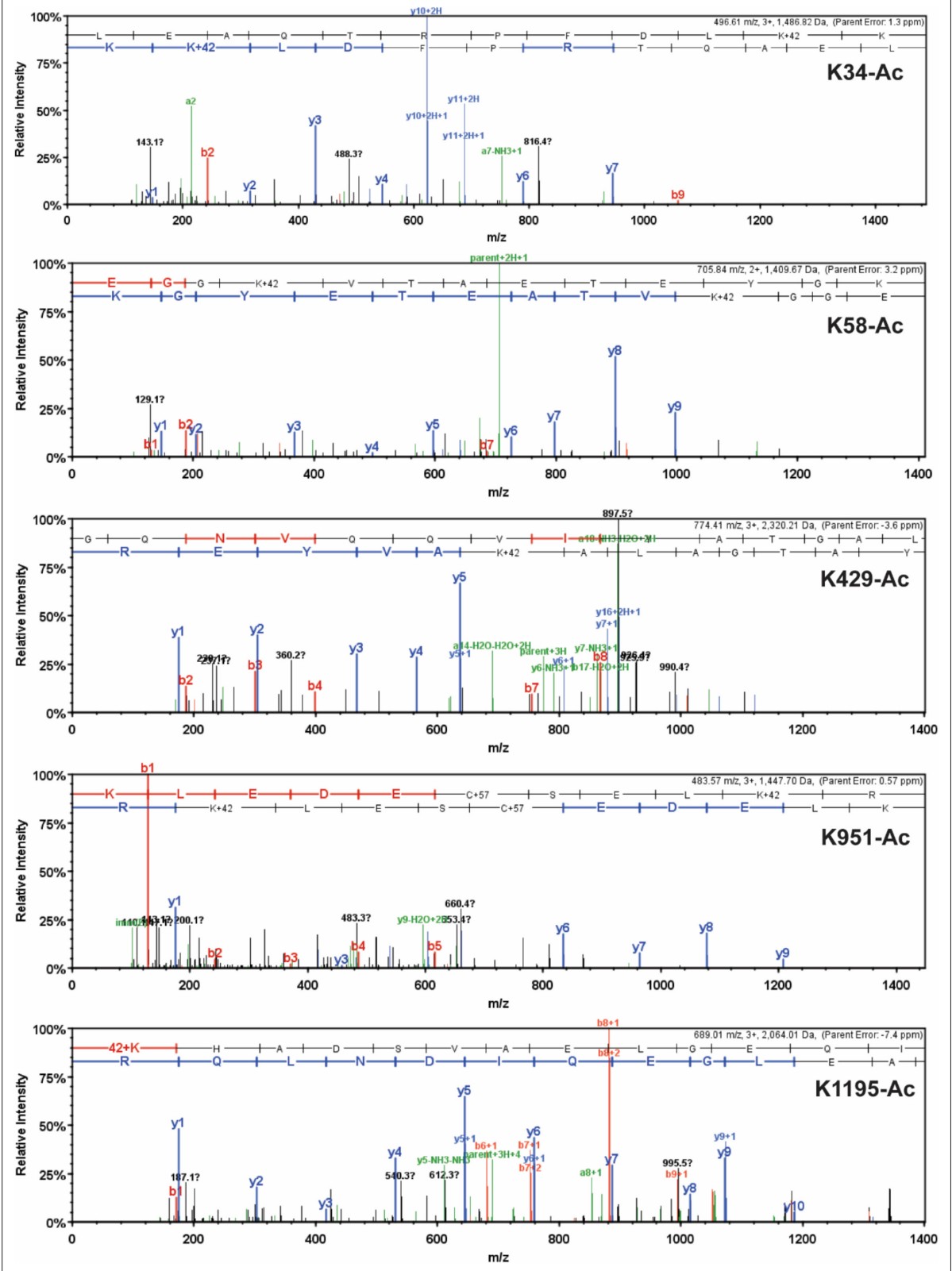

**Figure 2.** Detailed mass spectrometry (MS)/MS spectra of acetylated human β-myosin heavy chain (β-MHC) peptide sequences. MS/MS spectra of trypsin-digested, acetylated β-MHC peptide sequences 24–35 (K34-Ac, *m/z* 496.61), 55–67 (K58-Ac, *m/z* 705.84), 414–434 (K429-Ac, *m/z* 774.41), 942–952 (K951-Ac, *m/z* 483.57), and 1195–1212 (K1195-Ac, *m/z* 689.01). b-ions and y-ions indicate N-terminal and C-terminal ions, respectively.

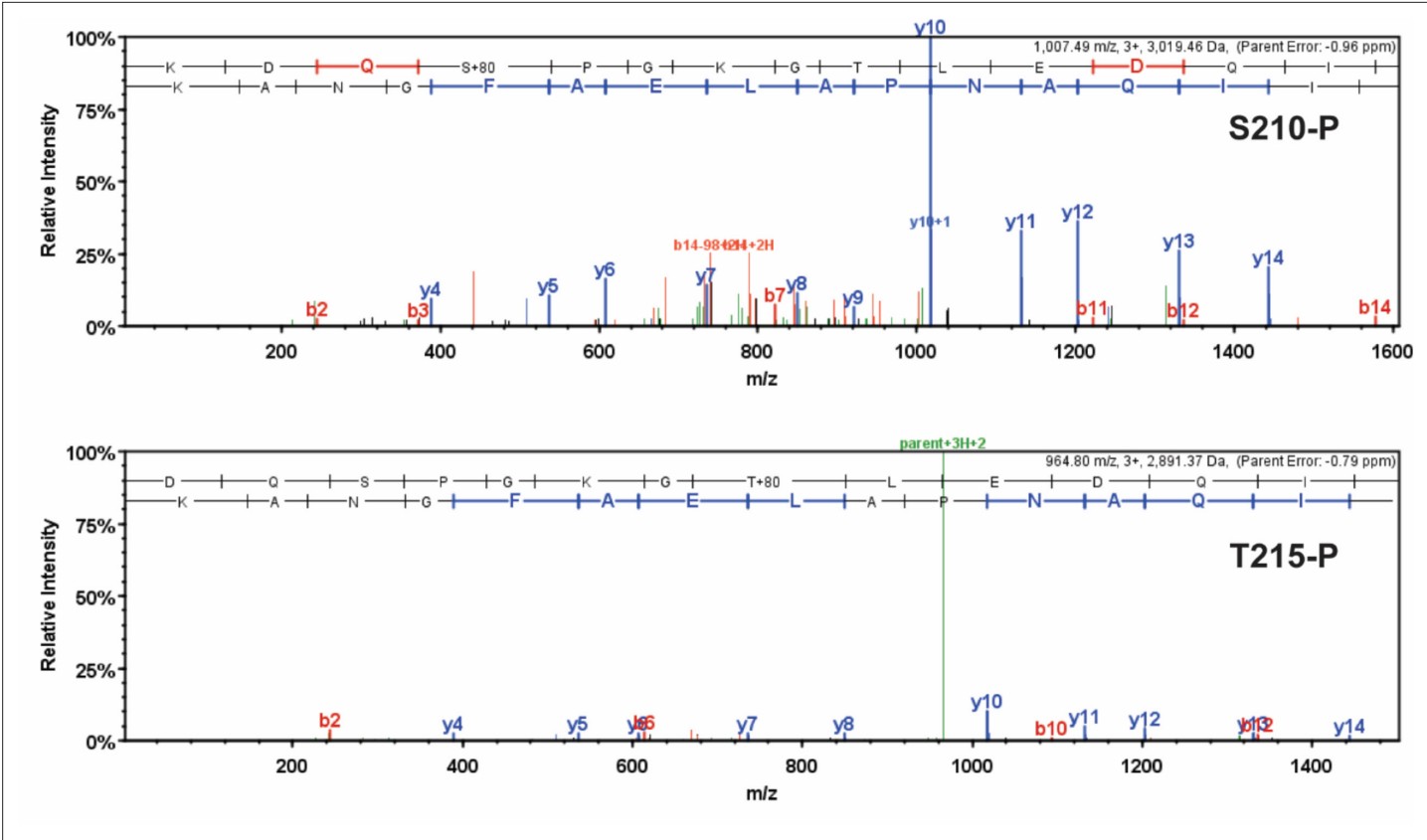

**Figure 3.** Detailed mass spectrometry (MS/MS) spectra of phosphorylated human β-myosin heavy chain (β-MHC) peptide sequences. MS/MS specta of trypsin-digested, phosphorylated β-MHC peptide sequences 207–234 (S210-P, *m/z* 1007.49) and 208–234 (T215-P, *m/z* 964.80). b-ions and y-ions indicate N-terminal and C-terminal ions, respectively.

low-abundance PTM with ratios of modified peptides/IRP ratios of approximately 1 and 1.4. This site is buried within the actin-binding cleft and may have been modified prior to protein folding. K34-Ac, on the other hand, is more abundant with approximate modified peptide ratios of 5–10. Perhaps the most interesting observation can be made for the co-modified peptide containing K58-Ac as it may be more susceptible to removal by HDACs or lower HAT activity under I-HF conditions.

In *Figure 6*, the modified/IRP ratios are shown for the phosphorylated peptides. The average ratios of T215-P ranged from 3 to 2, while in the control nondiseased hearts the modification was nearly undetectable in two samples. The modifications on the other high-confidence peptides may have functional significance, although the ratio of significantly modified/IRC was not altered in diseased (I-HF and NI-HF) compared to NF hearts (*Figures 5 and 6*). For this reason, we expect they may have a limited role in driving pathogenesis, but may still have functional relevance. Another phosphorylated residue was the phospho-serine at S210 on peptide 6, with ratios of approximately 1–2 modified peptide/IRP in all groups. T215 on peptide 7 had ratios of approximately 1–2 of the modified peptides/IRP in all the groups (*Figure 6*). When examining the 3D structure of the β-MHC motor domain, we observed three modifications – at the lateral face of the myosin motor domain – with potential importance due to their proximity to the ATP-binding pocket: (1) S210-P (single), (2) T215-P (single), and (3) K213-Ac plus T215-P (double). However, the question remains whether PTMs with low occupancy on a protein such as β-MHC may in fact have subtle yet important roles in fine-tuning of its function. Additional factors to consider are whether specific PTM sites reported here are more labile and prone to action by phosphatases and deacetylases, which may have increased activity in the failing heart.

**Table 1.** Liquid chromatography-mass spectrometry (LC-MS/MS) analysis of trypsin-digested human β-myosin heavy chain (β-MHC) sequences containing acetylated residues.

*An isotope dot product (idotp) value below 0.5 denotes a detectable but not quantifiable peptide sequence. $\underline{C}$ = carbamidomethyl. Acetylated residues are highlighted in red while phosphorylated residues are in blue.

| Protein | Peptide sequence | Protease | Modified residues | Modification | Calculated molecular mmass | Observed m/z | Observed molecular mass | Error (ppm) | Charge | .idotp |
|---|---|---|---|---|---|---|---|---|---|---|
| β-MHC | 23(R)LEAQTRPFDLKK(D)36 | Trypsin | K34 | Acetylation (+42) | 1487.82 | 496.61 | 1486.82 | 1.3 | 3+ | 0.99 |
| | 54(R)EGGKVTAETEYGK(T)68 | Trypsin | K58 | Acetylation (+42) | 1410.67 | 705.84 | 1409.67 | 3.2 | 2+ | 0.91 |
| | 206(K)DQSPGKGTLEDQIIQANPALEAFGNAK(T)235 | Trypsin | K213/T215 | Acetylation (+42) Phosphorylation (+80) | 3061.48 | 766.37 | 3061.48 | 0.0 | 4+ | 0.48* |
| | 413(K)GQNVQQVIYATGALAKAVYER(M)435 | Trypsin | K429 | Acetylation (+42) | 2321.22 | 774.41 | 2320.21 | −3.6 | 3+ | 0.93 |
| | 941(R)KLEDECSELKR(D)953 | Trypsin | K951 | Carbamidomethyl (+57) Acetylation (+42) | 1448.70 | 483.57 | 1447.69 | 0.57 | 3+ | 0.70 |
| | 1194(K)KHADSVAELGEQIDNLQR(V)1213 | Trypsin | K1195 | Acetylation (+42) | 2065.03 | 689.01 | 2064.01 | −7.4 | 3+ | 0.95 |

**Table 2.** Liquid chromatography-mass spectrometry (LC-MS/MS) analysis of trypsin-digested human β-myosin heavy chain (β-MHC) sequences containing phosphorylated residues.

*An isotope dot product (idotp) value below 0.5 denotes a detectable but not quantifiable peptide sequence. C = carbamidomethyl. Acetylated residues are highlighted in red while phosphorylated residues are in blue.

| Protein | Peptide sequence | Protease | Modified residues | Modification | Calculated molecular mass | Observed m/z | Observed molecular mass | Error (ppm) | Charge | idotp |
|---|---|---|---|---|---|---|---|---|---|---|
| β-MHC | $_{206}$(K)KDQSPGKGTLEDQIIQANPALEAFGNAK(T)$_{235}$ | Trypsin | S210 | Phosphorylation (+80) | 3020.47 | 1007.49 | 3019.46 | -0.96 | 3+ | 0.99 |
| | $_{206}$(K)KDQSPGKGTLEDQIIQANPALEAFGNAK(T)$_{235}$ | Trypsin | K213/T215 | Acetylation (+42) Phosphorylation (+80) | 3061.48 | 766.37 | 3061.48 | 0.0 | 4+ | 0.48* |
| | $_{207}$(K)DQSPGKGTLEDQIIQANPALEAFGNAK(T)$_{235}$ | Trypsin | T215 | Phosphorylation (+80) | 2892.37 | 964.80 | 2891.37 | -0.79 | 3+ | 0.99 |

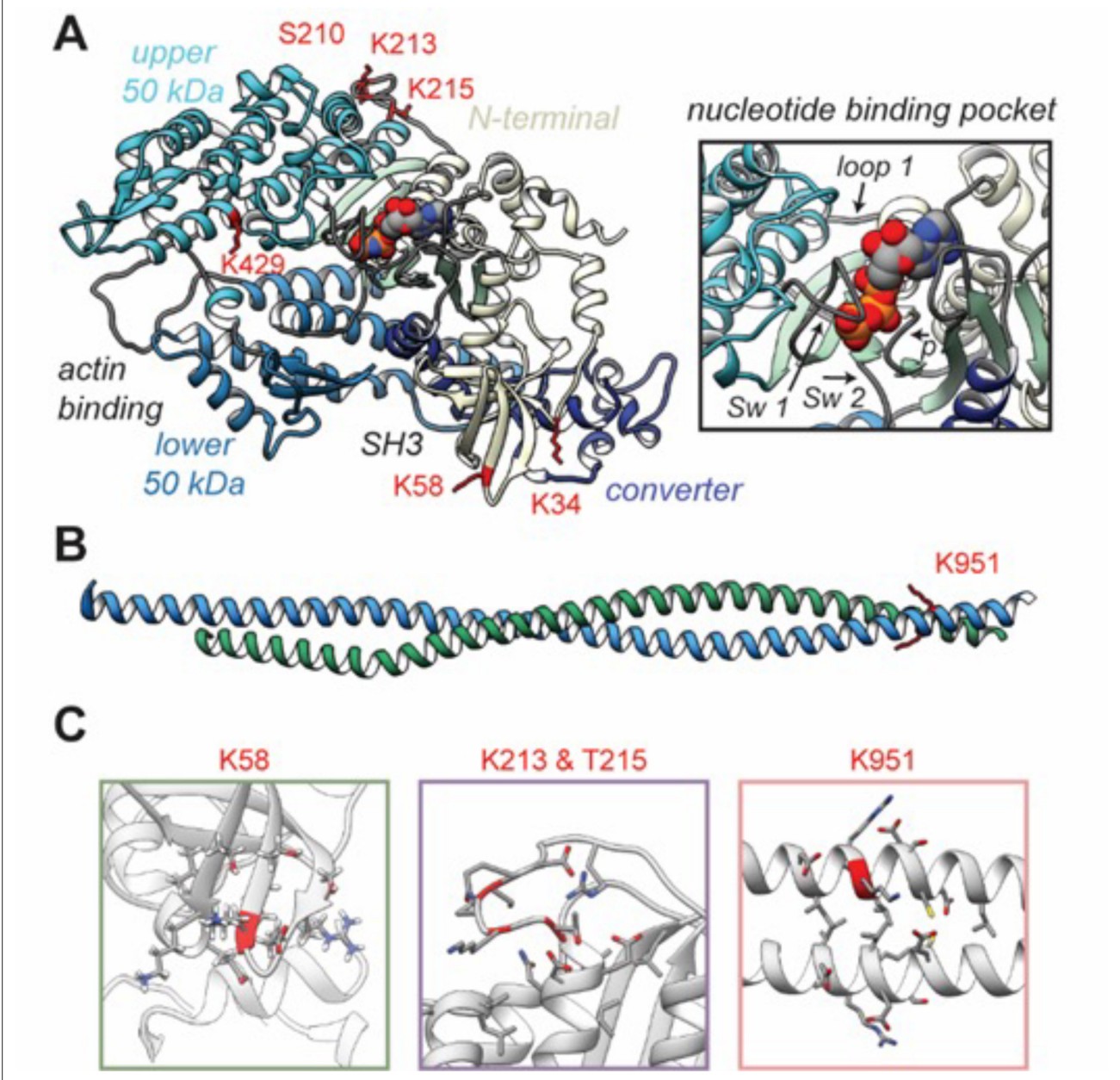

**Figure 4.** Structural models of post-translational modifications (PTMs) on β-myosin heavy chain (β-MHC). X-ray crystal structures of the β-MHC. In (**A**) a post-rigor X-ray structure was used to model K58-Ac, K213-Ac, T215-P PTMs. The four motor subdomains – the N-terminal domain (yellow), upper 50 kDa domain (cyan), lower 50 kDa domain (light blue), and converter domain (dark blue) – and function sites they form are labeled. Inset highlights the nucleotide-binding pocket and functional loops (loop 1, Switch 1, Switch 2, phosphate-binding loop). (**B**) An X-ray structure of an S2 fragment was used to model the K951-Ac PTM and served as the initial conformations of molecular dynamics (MD) simulations. In (**A, B**), residues with reported PTMs are shown and colored red. (**C**) The colored boxes display side-chain atoms in the vicinity of the modified (red ribbon) residues for K58 (green), K213/T215 (purple), and K951 (pink).

The online version of this article includes the following figure supplement(s) for figure 4:

**Figure supplement 1.** Detailed liquid chromatography-mass spectrometry (MS) spectrum of the common internal peptide sequence (IRP).

**Figure supplement 2.** Key functional regions of cardiac β-myosin motor domain and locations of the identified post-translational modifications (PTMs).

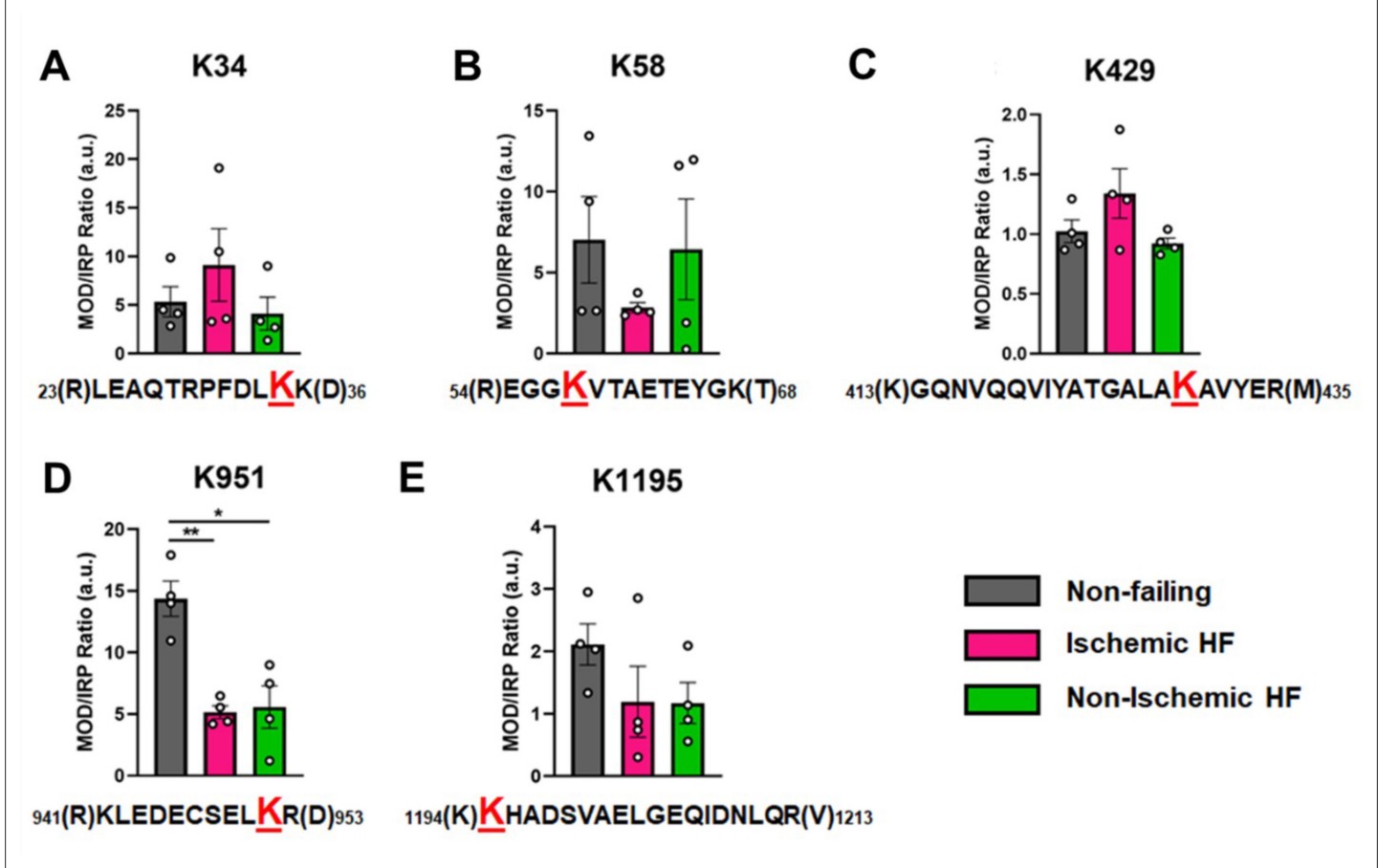

**Figure 5.** Calculations of post-translational modification (PTM) occupancy of acetylated residues on β-myosin heavy chain (β-MHC). Peak areas of all modified peptide sequences (MOD) were normalized to the peak area of a common internal reference peptide sequence (IRP, 1504–1521). (**A**) Peptide 1 sequence is shown with the site of acetylated lysine residue K34 indicated in red. (**B**) Peptide 2 is shown with the site of acetylated lysine residue K58 indicated in red. (**C**) Peptide 3 sequence is shown with the acetylated lysine residue K429. (**D**) Peptide 4 sequence is shown with the acetylated lysine residue K951 in red. (**E**) Peptide 5 sequence is shown with the acetylated lysine residue K1195 indicated in red. In the histograms, different human heart samples are indicated with nonfailing donor hearts (gray), ischemia-induced heart failure (pink), and nonischemia-induced heart failure (green). Trypsin-cutting sites are shown between parentheses. Data are expressed as mean ± SEM. Statistical analysis was performed by one-way ANOVA, *p<0.05, n = 4.

The online version of this article includes the following figure supplement(s) for figure 5:

**Figure supplement 1.** Mass spectrometry (MS/MS) spectrum of the doubly modified peptide sequence.

## Modeling the potential functional changes in β-MHC due to identified PTMs

Model building and MD were used to study putative relationships between PTMs identified on β-MHC and cardiovascular disease. Here, we performed MD simulations of unmodified and modified β-MHC (K58-Ac, K213-Ac/T215-P, K951-Ac) of PTMs that existed in functionally significant regions of the protein seen in *Figure 4*. For details regarding simulation runs, refer to *Tables 3 and 4*. From *Figure 4*, it can be seen that phosphorylation and acetylation sites were distributed throughout the structure of the motor domain and S2 fragment.

### Lysine 951

Lys 951 is located in S2 within the myosin tail (*Figure 4*). In mature myosin, the tail forms a coiled-coil structure where the helical tails of two MHCs are wrapped around one another. Archetypal coiled-coil helices have a conserved sequence repeat of seven amino acids: positions *a–g* in which positions *a* and *d* are hydrophobic residues that form a 'knobs in holes' interlocking structure that promotes

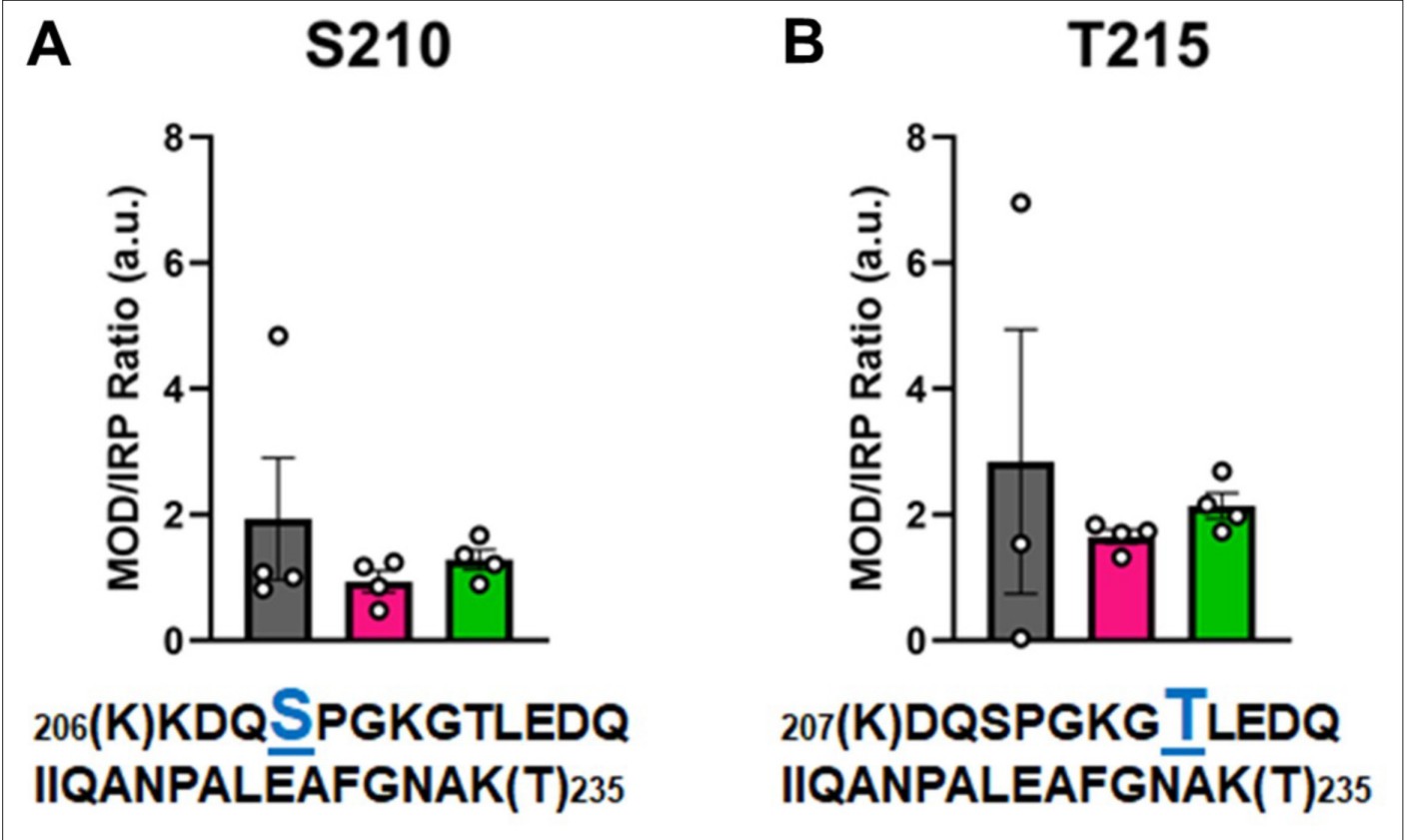

**Figure 6.** Calculations of post-translational modification (PTM) occupancy of phosphorylated residues on β-myosin heavy chain (β-MHC). Peak areas of all modified peptide sequences (MOD) were normalized to the peak area of a common internal reference peptide sequence (IRP, 1504–1521). (**A**) Peptide 1 sequence is shown with phosphorylated serine indicated in blue. (**B**) Peptide 2 sequence is shown with phosphorylated threonine shown in blue. Trypsin-cutting sites are shown between parentheses. Data are expressed as mean ± SEM. Statistical analysis was performed by one-way ANOVA, n = 3–4.

a well-packed hydrophobic core (*Truebestein and Leonard, 2016*). Myosin tails generally follow this archetypal pattern, but there are a small number of 'skip' residues that intermittently disrupt the heptad repeat pattern and increase local flexibility into the tail by weakening interactions in the hydrophobic core of the thick filament (*Taylor et al., 2015*). In implicit solvent MD, the simulated S2 fragment was flexible (*Table 4*), larger magnitude $C_\alpha$ RMSDs were associated with bending of the tail. Although bending occurred, the coiled-coil structure was largely preserved in the unmodified simulations (*Figure 7A*). Some coiled-coil structure was lost in the K951-Ac simulations in the vicinity of the modification site (*Figure 7B*). In both modified and unmodified simulations, there was partial

**Table 3.** Inventory of β-myosin heavy chain (β-MHC) post-translational modification (PTM) simulations.

Each row corresponds to a simulated system and reports the modifications that were made and the extent of molecular dynamics (MD) sampling.

| ID | PDB | Condition | Runs | Length per run (ns) | Net sampling (ns) |
|---|---|---|---|---|---|
| 4DB1 unmodified | 4DB1 | Unmodified | 3 | 500 | 1500 |
| 4DB1 K58-Ac | 4DB1 | K58Ac | 3 | 500 | 1500 |
| 4DB1 K213-Ac/T215-P | 4DB1 | K213Ac, T215P | 3 | 500 | 1500 |
| 2FXM K951 | 2FXM | Unmodified | 3 | 500 | 1500 |
| 2FXM K951-Ac | 2FXM | K951Ac | 3 | 500 | 1500 |

**Table 4.** Average $C_\alpha$ root mean squared deviation (RMSD) values.

The $C_\alpha$ RMSDs of each molecular dynamics (MD) snapshot versus the reference structure were averaged together per simulation.

| ID | Run number | $C_\alpha$ RMSD (Å) |
|---|---|---|
| 4DB1 unmodified | 1 | 3.9 ± 0.3 |
| | 2 | 3.2 ± 0.3 |
| | 3 | 4.1 ± 0.3 |
| 4DB1 K58-Ac | 1 | 3.8 ± 0.3 |
| | 2 | 3.6 ± 0.2 |
| | 3 | 4.1 ± 0.5 |
| 4DB1 K213-Ac/T215-P | 1 | 3.5 ± 0.4 |
| | 2 | 3.6 ± 0.4 |
| | 3 | 3.6 ± 0.3 |
| 2FXM unmodified | 1 | 6.9 ± 2.4 |
| | 2 | 7.5 ± 2.3 |
| | 3 | 7.2 ± 2.4 |
| 2FMX K951-Ac | 1 | 10.8 ± 2.8 |
| | 2 | 9.2 ± 2.5 |
| | 3 | 10.1 ± 2.0 |

The online version of this article includes the following source data for table 4:

**Source data 1.** Source data for *Table 4*.

unfolding at the C-terminal end of the fragment. This is likely due to the truncation of the structure or the exclusion of the crystallographically present Hg atoms. In simulations without PTMs, K951 formed transient interactions with E949 of the opposite chain, but most interactions made by K951 were hydrophobic and involved L950 of the opposite chain (*Figure 7A and C*). Acetylation of K951 had three effects on the structure of the tail. First, contacts made by K951 were altered: the frequency of salt bridge formation with Glu residues was diminished and K951-Ac interacted more frequently with other residues in the same chain as opposed to the opposite chain (*Figure 7B and C*). Second, the C-terminal end of the helix became bent and deviated from the typical coiled-coil structure. Third, the interhelical distance (measured by calculating the distance between $C_\alpha$ pairs in the two helices) increased in the presence of the PTM (*Figure 7D and E*). Additionally, K951-Ac altered the electrostatic potential of the S2 fragment, and notably the affected region is proximal to the interacting myosin heads in the super-relaxed conformation (*Figure 7F*). K951-Ac subverted the local coiled-coil structure of S2.

## Lysine 58

The overall motor domain in the K58-Ac simulations sampled conformations similar to those sampled by the WT simulations: the K58-Ac simulations had similar average $C_\alpha$ RMSD to the unmodified simulations (*Table 4*). We examined changes in structure of the SH3-like domain caused by K58-Ac. SH3 domains are typically modules of protein–protein interaction and canonically bind to proline-rich sequence with polyproline helical structure (*Kurochkina and Guha, 2012*). Putative poly-Pro-binding pockets are formed by K58 and K72 as well as K72, T70, T60, S53, and E55 found on the surface of MYH7's SH3 domain (*Figure 8A and C*). There is no direct structural evidence that these residues do form binding pockets; we have inferred this structural role from homology to other SH3 domains. In the unmodified simulations, both putative binding pockets remained intact and accessible to potential binding partners. In the K58-Ac simulations, however, the uncharged, acetylated Lys alternated between two conformations: a 'native-like' conformation in which the side chain projected into solvent and an 'intercalated' conformation in which the aliphatic portion of K58 was sandwiched in between

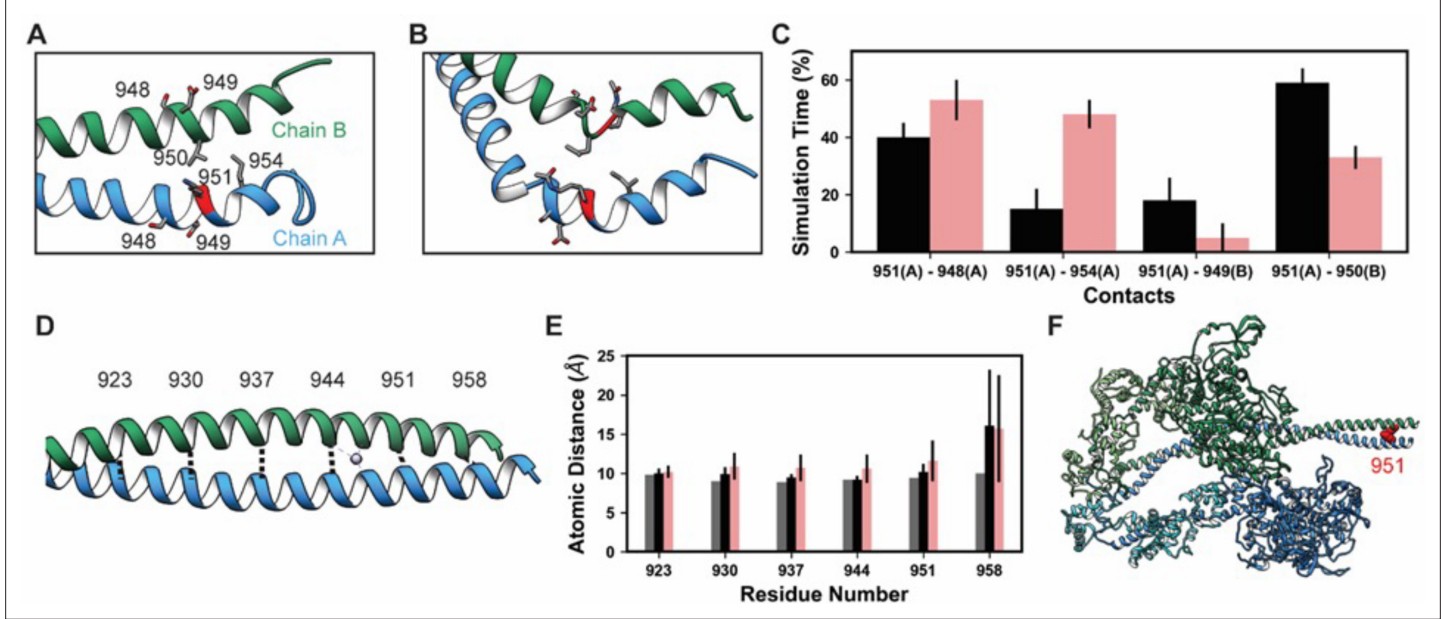

**Figure 7.** K951-Ac increased flexibility of the myosin tail. Structural changes in the S2 fragment caused by K951-Ac are shown in representative snapshots from molecular dynamics (MD) simulations. In the snapshots, the ribbon of residue 951 is colored red, chain A is colored blue, and chain B green. The atoms of neighboring side chains are displayed. The coiled-coil structure was preserved in the unmodified simulations (**A**). In the modified simulations (**B**), the coiled-coil structure was interrupted by kinks, a loss of α-helix structure, and increased separation of the chains. Collectively, these changes increased the local flexibility of S2. (**C**) Changes in S2 structure and dynamics caused by K951-Ac were associated with changes in local inter- and intra-chain contacts. (**D**) The inter-chain distance was monitored at six positions along S2, chosen to align with the heptad repeat position of the modified residue position. Distances between the $C_\alpha$ atoms between these residues were tracked for the X-ray structure (shown) and the MD simulations. (**E**) K951-Ac increased the inter-helix distance (~1 Å) relative to the unmodified simulations. This effect propagated towards the N-terminal end of the helix. The effect may also propagate towards the C-terminus; however, the structure is truncated at residue 961 and unfolding of the helices occurred in these simulations. (**F**) In the super-relaxed conformation, the portion of S2 affected by this modification is located nearby the motor domains as indicated on this model (PDB ID: 5TBY).

The online version of this article includes the following source data for figure 7:

**Source data 1.** Source data for *Figure 7*.

K72 and T70 (*Figure 8A and C*). The 'intercalated' conformation was associated with a disruption of the native contact network among SH3 residues, an increase in the solvent-accessible surface area (SASA) of residue K58 (partially attributable to the intrinsic increase in SASA of acetylated Lys), and a decrease in the SASA of K72 (*Figure 7B*). Importantly the 'intercalated' conformation abolished one of the putative poly-Pro-binding pockets on the surface of SH3 (*Figure 7C*), suggesting that K58 acetylation impedes binding of the SH3 domain to its targets.

### Lysine 213/threonine 215

Lys 213 and Thr 215 are both located in loop 1 of myosin S1, which is comprised of residues 199–215 in β-MHC (*Figure 9*). Loop 1 connects two α-helices that line the nucleotide-binding pocket. These helices in turn are connected to two loops that coordinate the contents of the nucleotide-binding pocket: switch 1 and the phosphate-binding loop. Loop 1 is flexible and is rarely resolved in X-ray structures. Residues 205–211 are not present in the 4DB1 X-ray crystal structure and thus were built into our structural model prior to performing our simulations. In the unmodified simulations of PTMs, T215 retained its crystallographic role as a helix capping residue and the crystallographic interactions between T215, D218, and Q219 were preserved. K213 formed a transient salt bridge with D337, transient hydrogen bonds with N334, and transient hydrophobic interactions with V338 (*Figure 9A*). These interactions tethered the C-terminal end of loop 1 to the upper 50 kDa domain. Residue–residue interactions were altered in the K213-Ac/T215-P simulations. K213 lost interactions with N334/D337 in the upper 50 kDa domain and instead formed interactions with T215. T215-P gained interactions with R204 and K206. The altered amino acid interactions caused loop 1 to sample from a distinct

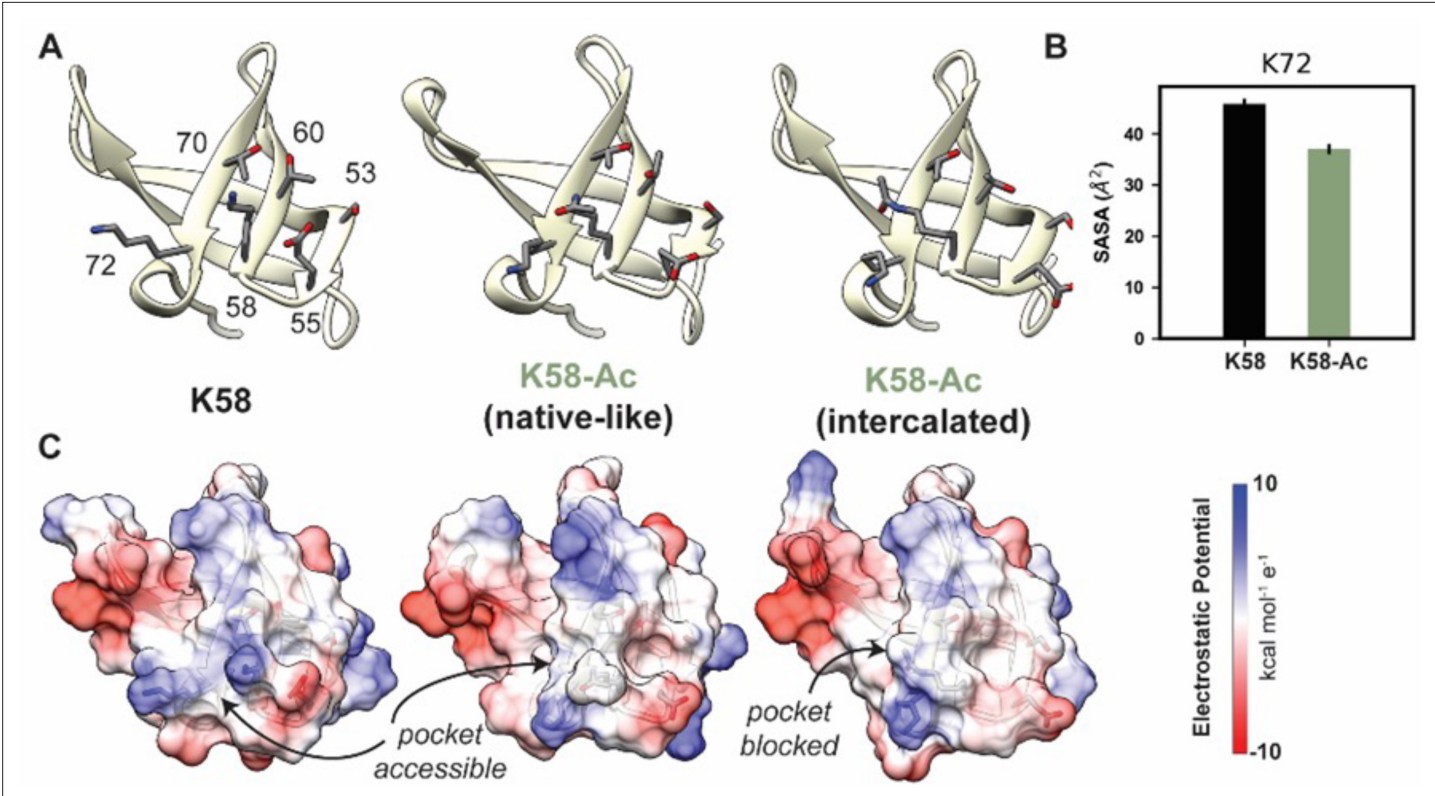

**Figure 8.** K58-Ac altered the solvent-exposed surface of myosin's SH3-like domain. (**A**) Representative structural changes in the SH3 domain associated with K58-Ac are shown in the endpoint structures of one unmodified ('K58') and two modified ('K58-Ac') simulations. K58-Ac formed increased interactions with T70, V71, and K72 in the neighboring strand and transiently formed an 'intercalated' conformation where the side chain was inserted between T70 and K72. (**B**) The transiently formed intercalated conformation led to a decrease in the solvent-accessible surface of K72. (**C**) Modification of K58 also altered the electrostatic potential of the SH3 domain surface, and in the intercalated conformation, K58-Ac blocks a surface pocket. The molecular surfaces in (**C**) correspond to the same structures shown in (**A**), and the electrostatic potential was calculated using *Chimera's* Coulombic surface coloring method.

The online version of this article includes the following source data for figure 8:

**Source data 1.** Source data for *Figure 8*.

conformational ensemble in the PTM simulations: it formed a more compact structure that involved more interactions with other loop 1 residues as opposed to an extended conformation with interactions to other regions of myosin.

## Discussion

Our study shows that the identified phosphorylation and acetylation PTMs are present on β-MHC in nondiseased human hearts. The presence of PTMs on β-MHC may fine-tune its function, altering the dynamics of some myosin motors in the cardiac sarcomere. Cumulatively, the functional alteration of a select number of β-MHC molecules may be a 'direct readout' of the cardiac cellular environment that dictates the needed changes in PTM addition/removal in context with degree of pathological insult and remodeling. Therefore, the presence of PTMs on cardiac β-MHC in nondiseased hearts suggests they may contribute to the normal functional and contractile state of the protein and muscle filaments that contain them.

MD simulations were performed to better understand the functional significance of the β-MHC PTMs. Focus was placed on modifications in modified residues with significantly altered abundance in diseased states: K951-Ac within S2 as well as for modified residues in regions with greatest functional significance: K58-Ac within the SH3-like domain, and K213-Ac and T215-P near the ATP-binding pocket.

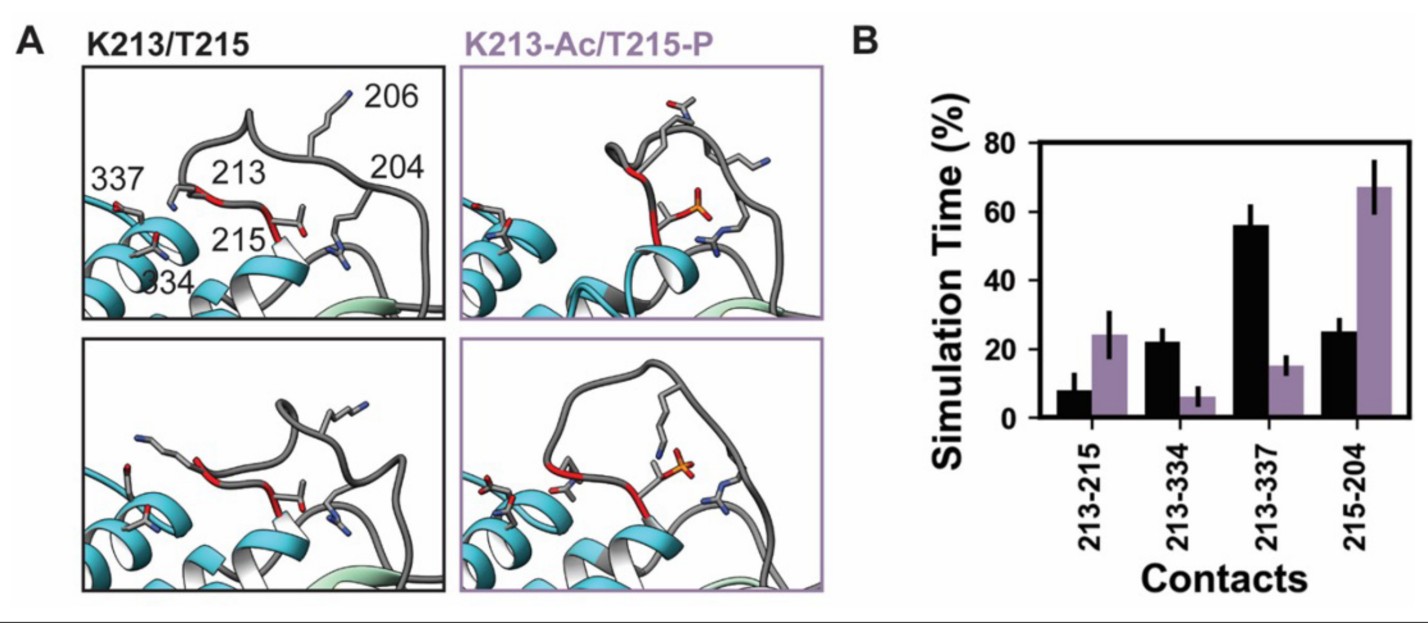

**Figure 9.** K58-Ac/T215-P altered loop 1 structure and dynamics. Molecular dynamics (MD) snapshots in (**A**) show representative structures from the unmodified (K213/T215) and modified (K213-Ac/T215-P) simulations. In the unmodified simulations, the loop makes long-lasting interactions with the upper 50 kDa domain (teal) of myosin. In the modified simulations, it became more compact and made fewer interactions with the upper 50 kDa domain. (**B**) The K213-Ac and T215-P post-translational modification (PTMs) altered the duration of inter-residue contacts made by K213 and T215: fewer interactions were made with the upper 50 kDa domain of myosin and more enduring interactions were made with other loop 1 residues.

The online version of this article includes the following source data for figure 9:

**Source data 1.** Source data for *Figure 9* (contacts).

**Source data 2.** Source data for *Figure 9* (distances).

## Acetylation of β-MHC

### Lysine 951

It is reasonable to propose that the observed reduction in acetylation at K951 may be associated with declining myocardial function. Our simulations suggest that acetylation of K951 disrupts the native structure of the coiled-coil and increases flexibility within the tail. Increased tail flexibility may alter the orientation of myosin heads relative to the thin filament, alter association rates between the thick and thin filament, and may affect the ability of myosin to form the interacting heads motif. Our simulations alone are not able to assess which of these effects on myosin dynamics is most likely due to the limited fragment of myosin that was simulated. Specifically, effects of these modifications on the flexibility and of S2 in general as well as on interactions between the myosin heads and S2 cannot be directly derived from these simulations. Nevertheless, our simulations do provide strong evidence for increased local flexibility in the tail. We attribute this to altered electrostatic interactions in the vicinity of K951.

### Lysine 1195

Acetylation of K1195 may result in similar effects. PTM of this region may further modulate tail flexibility or the assembly of the myosin rod (*Taylor et al., 2015*).

In our study, we see distinct differences in β-MHC acetylation that seem to be influenced more by location of the lysine than the type of heart disease (I-HF and NI-HF). The rod region is perhaps the more conformationally consistent portion of the myosin protein that is accessible to HATs but also HDACs. Therefore, one interpretation of the tendency for lower acetylation of K951 ($p<0.05$) and K1195 (not significant) in the failing heart samples is that they are not in the globular domain and may be more accessible to HDACs. It is unknown whether HDAC-inhibition would preserve acetylation at these sites and whether maintenance of this PTM is functionally important. Recent studies have explored the functional significance of sarcomeric protein acetylation to understand the impact

of HDAC on cardiac function (*Jeong et al., 2018*). The levels of acetylated proteins in the heart examined in this study depend on HDAC activity as much as HAT activity. Class I HDACs promote pathological cardiac hypertrophy, while class IIa HDACs suppress cardiac hypertrophy (*Zhang et al., 2012*). Interestingly, class II HDAC4 has been shown to be associated with the sarcomere (*Samant et al., 2015*; *Gupta et al., 2008*). Increased cardiac $Ca^{2+}$/calmodulin-dependent kinase II (CaMKII) expression and activity plays a role in heart failure development and progression by increasing class I HDAC activity (*Anderson et al., 2011*; *Zhang et al., 2020*). Administration of the HDAC inhibitor ITF2357 (Givinostat) improved heart relaxation in heart failure in rodent models with preserved ejection fraction (HFpEF) by promoting myofibril relaxation (*Jeong et al., 2018*). HDAC inhibitors are beneficial to cardiac function by targeting cytosolic and sarcomeric proteins by improving muscle contractility (*Demos-Davies et al., 2014*) and relaxation (*Jeong et al., 2018*) as well as cardioprotection during ischemia/reperfusion injury by promoting autophagy (*Xie et al., 2021*). It remains unknown whether reduced PTM abundance at K951 accompanied and/or precipitated the functional decline in the nonischemic and ischemic failing hearts or rather was a secondary result of the altered cellular environment.

## Lysine 58

While the remaining simulated modifications did not have significantly altered abundance in diseased states, the proximity of these PTMs to critical functional sites in myosin merited investigation. K58 acetylation is markedly reduced in the I-HF hearts, and its localization near the SH3 domain may also be slightly more exposed, with potentially greater HDAC activity in ischemic hearts. The other acetylated residues K34 and K429 are located further into the interior of the myosin head and may be less affected by increased HDAC activity in diseased hearts. Preservation of acetylated lysines in β-MHC may be beneficial to normal heart function, and pathological conditions favoring greater HDAC activation may disrupt crucial constitutively acetylated residues hastening heart failure development. The K58-Ac simulations indicated that K58-Ac caused minimal structural perturbations overall compared to the nonmodified protein. Instead, the greatest impact of K58-Ac is the decrease in electrostatic potential and structure on the outer surface of the SH3 domain. Lowey et al. and others have provided evidence that an interaction between the ELC and actin is mediated by the SH3 domain of myosin S1 (*Lowey et al., 2007*, *Aydt et al., 2007*). In light of this, our simulations suggest that K58-Ac has the potential to impede interactions between the ELC and SH3-like domains, thereby reducing interaction between the ELC N-terminus of actin filaments and effectively increasing shortening velocity. It remains possible that acetylation could increase affinity of SH3 for the ELC via altered intermolecular interactions, but we find this the less likely scenario. We suggest that K58-Ac may provide a reversible means to decrease the electrostatic potential of the SH3 domain surface, which may alter its interactions/affinity for ELC and perhaps other sarcomeric regulatory proteins.

The doubly modified peptide K213-Ac/T215-P was deemed to contain high-confidence PTM sites; however, K213-Ac/T215-P were found to be detectable but not quantifiable. A set of studies on loop 1 sequences (*Decarreau et al., 2011*; *Sweeney et al., 1998*) demonstrated that the length and composition of loop 1 regulate ADP release: shorter loops were associated with decreased ATPase activity, sliding velocity in in vitro motility assays, and preferential binding of ADP relative to ATP. In our simulations, the modified loop 1 formed a more compact structure and we predict that these PTMs would effectively behave as shorter loops that stabilize ADP. However, our simulations were performed in a post-rigor-like ATP-bound, actin-free structure of myosin and have insufficient sampling to measure ATP affinity or ATP-binding/ADP-release rates. Additionally, the rules that govern the relationship between loop 1 and nucleotide binding have not been definitely established, which complicates in silico predictions made here. We speculate that these structural perturbations in the ATP-binding pocket could alter the ADP-release rate.

## Phosphorylation of MHC

Several studies have provided insight on how phosphorylation of cardiac sarcomeric proteins alters myofilament performance (*Kawai et al., 2017*; *Sadayappan et al., 2006*; *Wijnker et al., 2014*; *Kuster et al., 2012*; *Solaro and Kobayashi, 2011*; *Yamasaki et al., 2002*). Kawai et al. utilized LC-MS and found that the rod region of α-MHC (predominant myosin isoform in the adult mouse heart) was hypophosphorylated in the HCM-linked cTnC-A8V mouse model, leading to perturbed cross-bridge

kinetics (*Kawai et al., 2017*). Additionally, the Frank-Starling mechanism may in turn be modified to increase cardiac output under conditions of increased venous return (*Monasky et al., 2013*). Incidentally, phosphorylation of myofilament proteins by PKCβII and PKA influences length-dependent prolongation of heart muscle relaxation (*Monasky et al., 2010*; *Monasky et al., 2008*). Moreover, sarcomeric protein phosphorylation is regulated by reactive oxygen species (ROS), whereby oxidative stress tends to increase protein phosphorylation due to inhibition of protein phosphatase activation and stimulation of protein kinases (*Sumandea and Steinberg, 2011*). Crosstalk between PTMs may in fact provide a higher order of regulation, for example, same-site competition, structural changes in secondary sites that make it more accessible for modification by other PTMs, or direct modification of the modification of a secondary PTM (*Liddy et al., 2013*).

## Extrapolation of function of modified regions
### Comparing known cardiomyopathy-associated variants with identified PTM sites

To gain further insight on the potential functional significance of the PTMs identified on cardiac β-MHC, we compared our results with nearby known variants linked to cardiomyopathic diseases in humans. There are numerous pathogenic variants in human cardiac slowing β-MHC, the predominant isoform in the myocardial ventricles. *MYH7* variants are implicated in roughly one-third of diagnosed familial HCM cases (*Ho et al., 2018*) and 10% of familial DCM cases (*Kamisago et al., 2000*). In this study, we compared the impact of cardiomyopathic variants located in the same region as the PTMs we identified, for example, DCM-linked variants T412N in the head region and R1193S in the tail domain (*Villard et al., 2005*). These variants may impact actin–myosin interactions or impair myosin rod structure and assembly (*Villard et al., 2005*). Additionally, we show that the PTMs K58-Ac, S210-P/T215-P, and K429-Ac are located in close proximity to pathogenic variants and are present under nondiseased conditions and tend to decrease in heart failure in most cases with a significant decrease in K951-Ac. See *Supplementary file 2* for comparisons between location of PTMs found in this study with existing reports of pathogenic variants in the *MYH7* gene.

## Conclusions

Our study identified novel PTMs on β-MHC in nonfailing and failing human hearts. Overall, there tended to be less PTM abundance in the heart failure conditions examined here, which may be due to increased phosphatase or HDAC activity. Our modeling data suggest that some of these PTMS have potential to alter dynamics within β-MHC and thereby fine-tune myofilament function. It remains to be seen whether loss of these PTMs in failing hearts represents the removal of a beneficial, albeit subtle regulation of myofilament function.

## Materials and methods
### Human heart samples

Explanted donor human heart tissues were obtained from the Ohio State University Tissue program. The de-identified samples were obtained from patients that were 41–69 years of age with I-HF or NI-HF, and healthy donors after informed consent. Clinical information about patient hearts that were utilized in this study is included in *Supplementary file 1*.

### Heart sample preparation

Human heart tissue was homogenized in Laemmli buffer and separated by SDS-PAGE (12 %), stained with Coomassie blue, and bands that corresponded to β-MHC (~223 kD) were excised as shown in *Figure 1—figure supplement 1*. The samples were homogenized in 1× Laemmli sample buffer with protease inhibitor cocktail, phosphatase inhibitor cocktail, 1 μM trichostatin A, 1 μM quisinostat, 5 mM nicotinamide, and 1 mM sodium vanadate.

## Mass spectroscopy

### Sample preparation

In-gel digests were performed for each excised sample using the ProteoExtract All-in-One Trypsin Digestion Kit (Cat#. 650212; Calbiochem, EMD Millipore, Billerica, MA) according to the manufacturer's instructions. Briefly, excised gel pieces were destained in wash buffer and dried at 90°C for 15 min. Gel pieces were rehydrated in trypsin digestion buffer and treated with a reducing agent for 10 min at 37°C. Samples were cooled to room temperature and then incubated in blocking reagent for 10 min at room temperature. Trypsin was added to a final concentration of 8 ng/µl and incubated for 2 hr at 37°C on an orbital shaker. Peptides were eluted in 50 µl 0.1% formic acid.

### Liquid chromatography-mass spectrometry (LC-MS)

Sample peptides were processed using an externally calibrated high-resolution electrospray tandem mass spectrometer (Thermo Q Exactive HF; Thermo Fisher Scientific, Waltham, MA) in conjunction with a Dionex UltiMate 3000 RSLCnano System (Thermo Fisher Scientific). 5 µl of sample peptide was aspirated into a 50 µl loop and then loaded onto the trap column (Acclaim PepMap100 C18, 5 µm, 100 Å, 300 µm i.d. × 5 mm, Cat# 160454, Thermo Fisher Scientific). Separation on an analytical column (Acclaim PepMap RSLC 75 µm and 15 cm nanoViper; Thermo Fisher Scientific) was conducted with a flow rate of 300 nl/min. A 60 min linear gradient from 3% to 45% B (0.1% formic acid in acetonitrile) was performed. The LC eluent was directly nanosprayed into Q Exactive HF mass spectrometer (Thermo Scientific). During the chromatographic separation, the Q Exactive HF was operated in a data-dependent mode and under direct control of the Thermo Excalibur 3.1.66 (Thermo Scientific). The MS data were acquired using the following parameters: 20 data-dependent collisional-induced dissociation (CID) MS/MS scans per full scan (350–1700 $m/z$) at 60,000 resolution. MS2 were acquired in centroid mode at 15,000 resolution. Ions with single charge or charges more than 7 as well as unassigned charge were excluded. Raw data were searched with Proteome Discoverer 2.2 (Thermo Fisher Scientific) using Sequest HT and Mascot search engines and percolator as the PSM validator with species-specific FASTA database. Phosphorylation and acetylation were used as a dynamic modification in SequestHT and Mascot and PTMs were scored by ptmRS node in Proteome Discoverer 2.2. MS1-based quantification of peptides was performed using Skyline 4.0. The mass spectrometry data generated in this study has been deposited at Dryad and can be found at doi:10.5061/dryad. s4mw6m97g.

## Proteomics data analysis

The area under the curve (AUC) for each peptide sequence was measured in Skyline 20.1, an open-source Windows client application software for targeted proteomics data analysis and bioinformatics (*MacLean et al., 2010*). To determine the modification levels of each PTM site, the AUC of each modified peptide was measured and divided by the AUC of the unmodified common internal reference peptide (IRP) (*Figure 5—figure supplement 1*). The ratio of the modified peptide to IRP was compared among human heart samples from nondiseased controls and patients with I-HF and NI-HF. The IRP was chosen based upon recommendations from Sherrod et al. utilizing a known unmodified peptide between 7 and 20 amino acids (no methionine and cysteine residues), eluted across the chromatogram, and demonstrating consistent signal stability (*Sherrod et al., 2012*).

## Model building

Starting coordinates for the wildtype (WT) β-MHC motor domain simulations were obtained from an X-ray crystallography structure of the post-rigor, ATP state of β-MHC in the Protein Data Bank (PDB, https://www.rcsb.org; PDB ID: 4DB1, 2.6 Å, residues 1–777; *Berman et al., 2000*). Starting coordinates for the WT S2 fragment simulations were obtained from an X-ray crystallography structure of S2D (PDB ID: 2FXM, 2.7 Å, residues 838–963) (*Blankenfeldt et al., 2006*). Missing heavy atoms were built using *Modeller* (*Webb and Sali, 2016*), and conformer 'A' was chosen among residues with multiple conformations in the PDB entries. Hg atoms were removed from the 2FXM structure and ANP•Mn was replaced with ATP•Mg in 4DB1. Starting coordinates for the post-translationally modified variants were obtained via in silico modification of the WT structures using the *leap* module of *AMBER*. There were two modified variants of 4DB1: 4DB1-K58-Ac (corresponding to acetylation of Lys 58) and 4DB1-K213-Ac/T215-P (corresponding to simultaneous acetylation of Lys 213 and

phosphorylation of Thr 215). There was one modified variant of 2FXM: 2FXM-K951-Ac (corresponding to acetylation of Lys 951). For 2FXM, K951 residues in both chains A and B were modified. Because both K951 residues were modified, our simulations sample the 'most aggressive' effects of acetylation at this site: diminished effects may be found for singly modified systems.

## Force field and explicit solvent molecular mechanics

All 4DB1 simulations were performed with the AMBER20 package (*Geeves and Holmes, 1999*; *Tang et al., 2016*) and the ff14SB force field (*Maier et al., 2015*). Water molecules were treated with the TIP3P force field (*Sumandea and Steinberg, 2011*). Metal ions were modeled using the Li and Merz parameter set (*Li and Merz, 2014*; *Li et al., 2015a*; *Li et al., 2015b*). ATP molecules were treated with parameters from *Meagher et al., 2003*. Parameters for phosphothreonine (called 'TPO') and acetyl-lysine (called 'ALY') were obtained from Raguette et al. and Belfon et al., respectively. The SHAKE algorithm was used to constrain the motion of hydrogen-containing bonds (*Hammonds and Heyes, 2020*). Long-range electrostatic interactions were calculated using the particle mesh Ewald (PME) method (*Essmann et al., 1995*).

## Preproduction protocols

Hydrogen atoms were modeled onto the initial structure using the *leap* module of *AMBER,* and each protein was solvated with explicit water molecules in a periodic, truncated octahedral box that extended 10 Å beyond any protein atom. Na$^+$ and Cl$^-$ counterions were added to neutralize the systems and then 120 mM Na$^+$ and Cl$^-$ ions were added. Each system was minimized in three stages. First, hydrogen atoms were minimized for 1000 steps in the presence of 100 kcal/mol restraints on all heavy atoms. Second, all solvent atoms were minimized for 1000 steps in the presence of 25 kcal/mol restraints on all protein atoms. Third, all atoms were minimized for 8000 steps in the presence of 25 kcal/mol restraints on all backbone heavy atoms (N, O, C, and C atoms). After minimization, systems were heated to 310 K during three successive stages. In each stage, the system temperature is increased by ~100 K over 100 ps (50,000 steps) using the NVT (constant number of particles, volume, and temperature) ensemble. During all heating stages, 25 kcal/mol restraints were present on the backbone heavy atoms (N, O, C, and C atoms). After the system temperatures reached 310 K, the systems were equilibrated over five successive stages using the NPT (constant number of particles, pressure, and temperature) ensemble. During each stage, the systems were equilibrated for 5.4 ns in the presence of restraints on backbone atoms. The strength of the restraints was decreased from 25 kcal/mol during the first stage to 1 kcal/mol during the fourth stage. During the final equilibration stage, the systems were equilibrated in the absence of restraints.

## MD protocol

Production dynamics for conventional MD simulations were then performed using the canonical NVT ensemble with an 8 Å nonbonded cutoff and 2 fs time step. Coordinates were saved every picosecond. Simulations were run in triplicate for 500 ns each. Unless specified otherwise, simulations were analyzed separately, and the results of replicate simulations were averaged together. To account for potential equilibration effects, the first 100 ns were excluded from subsequent analyses.

## Implicit solvent simulations

2FXM contains a linear fragment of S2 spanning ~130 residues. Due to the length of this fragment, we performed implicit solvent simulations of 2FMX using the Generalized Born model. This approach was previously used to study S2 fragments, and our methods were chosen to best match earlier simulations (*Korkmaz et al., 2016*). Simulations were performed using the GB model described by *Mongan et al., 2007*.

## MD analysis

The C$_\alpha$ RMSD, C$_\alpha$ RMSF, SASA, interatomic distances, and interatomic contacts were calculated with *cpptraj* (*Roe and Cheatham, 2013*). The C$_\alpha$ RMSD was calculated after alignment of all The C$_\alpha$ atoms to the minimized structure. The C$_\alpha$ RMSF was calculated about average MD structures for each simulation. Two residues were considered in contact with one another if at least one pair of heavy atoms were within 5 Å of one another. All protein images were prepared using UCSF Chimera (*Pettersen*

*et al., 2004*; *Sanner et al., 1996*). Electrostatic potentials of molecular surfaces were generated using default parameters for the *Coulombic Surface coloring* method in *UCSF Chimera*.

## Statistical analysis

Mass spectrometry and MD simulations data analyses were performed using one-way ANOVA, followed by Bonferroni's post hoc test and Student's *t*-test, respectively.

## Acknowledgements

The authors thank the Lifeline of Ohio for the collaboration on nonfailing donor tissue, and surgeons and transplant coordinators at the Ohio State University Wexner Medical Center for helping obtain the end-stage failing tissue. Funding for MSP was provided by the American Heart Association Award # 16SDG2912000 and FSU CRC Planning Grant #46259. Funding for JRP was provided by NIH grant HL128683. Funding to ML-V was provided by the American Heart Association Pre-doctoral Award 2021AHAPRE216237. This work used the Extreme Science and Engineering Discovery Environment (XSEDE) resource COMET through allocation TG-MCB200100 to MCC and MR. XSEDE was supported by the National Science Foundation grant number ACI-1548562. Funding for MCC was provided by Award Number T32HL007828 from the National Heart, Lung, and Blood Institute. The content is solely the responsibility of the authors and does not necessarily represent the official view of the NHLBI or the NIH. This research was supported by the University of Washington Center for Translational Muscle Research (CTMR) via the National Institute of Arthritis and Musculoskeletal and Skin Diseases of the National Institutes of Health award number P30AR074990.

## Additional information

### Funding

| Funder | Grant reference number | Author |
| --- | --- | --- |
| American Heart Association | 16SDG2912000 | Michelle S Parvatiyar |
| Florida State University | 46259 | Michelle S Parvatiyar |
| National Institutes of Health | HL128683 | J Renato Pinto |
| American Heart Association | 2021AHAPRE216237 | Maicon Landim-Vieira |
| National Science Foundation | ACI-1548562 | Michael Regnier |
| National Institutes of Health | T32HL007828 | Matthew C Childers |
| National Institutes of Health | P30AR074990 | Michael Regnier |

The funders had no role in study design, data collection and interpretation, or the decision to submit the work for publication.

### Author contributions

Maicon Landim-Vieira, Conceptualization, Formal analysis, Investigation, Methodology, Validation, Writing – review and editing; Matthew C Childers, Formal analysis, Investigation, Methodology, Software, Validation, Writing – original draft, Writing – review and editing; Amanda L Wacker, Formal analysis, Investigation, Methodology, Validation; Michelle Rodriquez Garcia, Data curation, Investigation; Huan He, Data curation, Formal analysis, Validation; Rakesh Singh, Data curation, Formal analysis; Elizabeth A Brundage, Methodology; Jamie R Johnston, Conceptualization, Writing – review and editing; Bryan A Whitson, Resources; P Bryant Chase, Investigation, Methodology, Supervision, Writing – review and editing; Paul ML Janssen, Resources, Writing – review and editing; Michael Regnier, Investigation, Methodology, Writing – review and editing; Brandon J Biesiadecki,

Conceptualization, Investigation, Resources; J Renato Pinto, Conceptualization, Project administration, Resources, Writing – review and editing; Michelle S Parvatiyar, Conceptualization, Formal analysis, Investigation, Methodology, Project administration, Resources, Supervision, Writing – original draft, Writing – review and editing

## Author ORCIDs
Matthew C Childers ⓘ http://orcid.org/0000-0003-2440-9612
Amanda L Wacker ⓘ http://orcid.org/0000-0002-7580-7189
P Bryant Chase ⓘ http://orcid.org/0000-0001-9701-561X
J Renato Pinto ⓘ http://orcid.org/0000-0001-9092-4976
Michelle S Parvatiyar ⓘ http://orcid.org/0000-0002-9416-0069

## Ethics

Human subjects: This study was conducted with the highest ethical standards, human heart samples were collected and stored with full consent of parties involved and were provided by the Lifeline of Ohio with coordination from surgeons and transplant coordinators at the Ohio State University Wexner Medical Center. All aspects of this study were approved and conform to the ethical guidelines established by the Institutional Review Board of The Ohio State University under protocol #2012H0197.

## Decision letter and Author response

Decision letter https://doi.org/10.7554/eLife.74919.sa1
Author response https://doi.org/10.7554/eLife.74919.sa2

---

# Additional files

## Supplementary files

• Supplementary file 1. Summary of the patients' demographic features. Deidentified human heart samples were obtained from nonfailing, ischemic heart failure, and nonischemic heart failure patients.

• Supplementary file 2. Comparison of location of residues bearing post-translational modifications (PTMs) with known cardiomyopathy variants in MYH7. List of potential pathogenicity of the variants and their location within nearby PTM regions. Cardiomyopathy-loop (CM-loop), likely pathogenic (LP), pathogenic (P), hypertrophic cardiomyopathy (HCM), and dilated cardiomyopathy (DCM).

• Transparent reporting form

## Data availability

All data generated or analyzed during this study are included in the manuscript and the supporting files have been provided for Figures 2, 3, 7, 8 , 9 and Figure supplements 1, 2, 4, Tables 1, 2, 3 and 4. Mass spec data have been deposited at Dryad under the unique identifier DOI (doi:https://doi.org/10.5061/dryad.s4mw6m97g).

The following dataset was generated:

| Author(s) | Year | Dataset title | Dataset URL | Database and Identifier |
|---|---|---|---|---|
| Parvatiyar MS | 2022 | Data from: Post-translational modification patterns on β-myosin heavy chain are altered in ischemic and non-ischemic human hearts | http://dx.doi.org/10.5061/dryad.s4mw6m97g | Dryad Digital Repository, 10.5061/dryad.s4mw6m97g |

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
