## [Editor Report]

This article surveys differences in the heavy chain of the contractile protein β-myosin in normal hearts and hearts in cardiac failure. This is important in view of its possible regulatory roles in generating contraction. The findings are then substantiated by functional simulations of the contractile process.

---

## [Decision Letter]

**Decision letter after peer review:**

Thank you for submitting your article "Post-translational modification patterns on β-myosin heavy chain are altered in ischemic and non-ischemic human hearts" for consideration by *eLife*. Your article has been reviewed by 3 peer reviewers, one of whom is a member of our Board of Reviewing Editors, and the evaluation has been overseen by a Senior Editor. The following individual involved in review of your submission has agreed to reveal their identity: Stuart Campbell (Reviewer #3).

The reviewers have discussed their reviews with one another, and the Reviewing Editor has drafted this letter to help you prepare a revised submission.

Essential revisions:

(A) General alterations requested.

(1) Both reviewers 2 and 3 are positive about the paper but both indicate a need for shortening of the paper.

(2) In particular reviewer 2,

although:

(a) generally positive in stating that the paper (i) represents an enormous amount of work on identifying modifications of the major cardiac motor protein β, myosin in the failing and non-failing human heart, and (ii) tackles an important understudied problem of PTMs in the human β cardiac myosin protein.

However,

(b) flags up the point that much of the material might rather go to a more specialized journal. On discussion, reviewer 2 specifically requests clarification why the findings should be of interest to readers of the broader remit represented by *eLife*.

(3) Reviewer (2):

Considers that this is a thought-provoking study using an appealing strategy. Human samples increase the disease relevance and mass spectrometry here yields novel discoveries. Molecular dynamics modeling is used to speculate on the relative functional impact of each PTM. The novel PTM sites identified in this work are exciting and should stimulate and inspire other work in the field.

(B) Specific alterations requested:

1) Reviewer 2:

a) There is a lot of information in this very long (too long in my opinion) right now, its utility is unknown. The PTMs are very rare on this molecule and there are not enough samples to have much power (even though I know how hard it is to get samples).

b) The authors also make several statements about how MYH6 or α myosin is induced in human heart failure. To the best of my knowledge, the opposite is what happens in human heart failure.

2) Reviewer 3:

a) I found it very difficult to understand the way in which PTM abundance was presented. It was not clear what the overall percentage of PTM is for a given site, e.g. in a failing heart, what percentage of myosin K951 residues are acetylated? The first paragraph of the discussion states that "phosphorylation and acetylation PTMs are present on β-MHC in non-diseased human hearts in low abundance." Is it not possible to determine what fraction of a particular peptide is modified? Otherwise, why would the authors state that PTMs are "low abundance" (first paragraph of the discussion)? Furthermore, if this information has been obtained, why would it not be clearly represented in the manuscript? It seems as though the quantification of PTMs is shown only relative to IRP.

b) Similarly, the PTM to IRP ratio is also confusing in that the stated range is 1-14. Does this mean that it is not possible for PTM abundance to be less than the IRP? Why not?

c). The authors mention that one way to interpret the loss of K951 acetylation in failing hearts is that there is a shift toward α-MHC. Perhaps it was outside the scope of the study, but was there no way of testing this hypothesis in some way?

d). The discussion is, in my view, far too long and wide-ranging. Some of the logic presented is also challenging. For instance, the PTMs discussed are shown to be more abundant in normal than in failing hearts. Yet, the authors draw parallels between the sites for the PTMs and the locations of cardiomyopathy-causing mutations. It seems strange to associate sites of "normal" PTMs with the locations of disease-causing mutations. Overall, the discussion is far longer than would normally be expected for the corresponding significant results, which really focused just on K951. My recommendation would be to focus the discussion drastically, state the most meaningful findings very clearly, and help the reader grasp the real significance of the work.

*Reviewer #2 (Recommendations for the authors):*

This paper tackles an important understudied problem of PTMs in the human β cardiac myosin protein. The authors found acetylation sites and phosphorylation sites and used molecular dynamics simulations to predict the impacts of these PTMs on myosin function.

There is a lot of information in this very long (too long in my opinion) right now, its utility is unknown. The PTMs are very rare on this molecule and there are not enough samples to have much power (even though I know how hard it is to get samples).

The authors also make several statements about how MYH6 or α myosin is induced in human heart failure. To the best of my knowledge, the opposite is what happens in human heart failure.

I think this information may be better suited for a more specialized audience and journal.

*Reviewer #3 (Recommendations for the authors):*

This is a thought-provoking study that makes use of an appealing strategy. Human samples increase the disease relevance of the work, and mass spectrometry has been used here to yield novel discoveries. Molecular dynamics modeling is used to speculate on the relative functional impact of each PTM. The novel PTM sites identified in this work are exciting and should stimulate and inspire other work in the field.

Some suggestions for improving the work are given below.

1. I found it very difficult to understand the way in which PTM abundance was presented. It was not clear what the overall percentage of PTM is for a given site, e.g. in a failing heart, what percentage of myosin K951 residues are acetylated? The first paragraph of the discussion states that "phosphorylation and acetylation PTMs are present on β-MHC in non-diseased human hearts in low abundance." Is it not possible to determine what fraction of a particular peptide is modified? Otherwise, why would the authors state that PTMs are "low abundance" (first paragraph of the discussion)? Furthermore, if this information has been obtained, why would it not be clearly represented in the manuscript? It seems as though the quantification of PTMs is shown only relative to IRP.

2. Similarly, the PTM to IRP ratio is also confusing in that the stated range is 1-14. Does this mean that it is not possible for PTM abundance to be less than the IRP? Why not?

3. The authors mention that one way to interpret the loss of K951 acetylation in failing hearts is that there is a shift toward α-MHC. Perhaps it was outside the scope of the study, but was there no way of testing this hypothesis in some way?

4. The discussion is, in my view, far too long and wide-ranging. Some of the logic presented is also challenging. For instance, the PTMs discussed are shown to be more abundant in normal than in failing hearts. Yet, the authors draw parallels between the sites for the PTMs and the locations of cardiomyopathy-causing mutations. It seems strange to associate sites of "normal" PTMs with the locations of disease-causing mutations. Overall, the discussion is far longer than would normally be expected for the corresponding significant results, which really focused just on K951. My recommendation would be to focus the discussion drastically, state the most meaningful findings very clearly, and help the reader grasp the real significance of the work.

---

## [Author Response]

Essential revisions:(A) General alterations requested.(1) Both reviewers 2 and 3 are positive about the paper but both indicate a need for shortening of the paper.(2) In particular reviewer 2,although:(a) generally positive in stating that the paper (i) represents an enormous amount of work on identifying modifications of the major cardiac motor protein β, myosin in the failing and non-failing human heart, and (ii) tackles an important understudied problem of PTMs in the human β cardiac myosin protein.However,(b) flags up the point that much of the material might rather go to a more specialized journal. On discussion, reviewer 2 specifically requests clarification why the findings should be of interest to readers of the broader remit represented by eLife.

Please see our comments for Reviewer 2, pt. c.

(3) Reviewer (2):Considers that this is a thought-provoking study using an appealing strategy. Human samples increase the disease relevance and mass spectrometry here yields novel discoveries. Molecular dynamics modeling is used to speculate on the relative functional impact of each PTM. The novel PTM sites identified in this work are exciting and should stimulate and inspire other work in the field.

We thank the editors and the expert reviewers for giving us the opportunity to further improve our manuscript. We have made significant changes in the manuscript by carefully following the reviewers’ suggestions and comments. In addition, we clarified the reasons why we chose *eLife* to submit our exciting study and why the manuscript (in its present form) should engage a broad audience. New sentences (in blue) were added throughout results and discussion to improve the flow of the manuscript after it was substantially shortened.

(B) Specific alterations requested:1) Reviewer 2:a) There is a lot of information in this very long (too long in my opinion) right now, its utility is unknown. The PTMs are very rare on this molecule and there are not enough samples to have much power (even though I know how hard it is to get samples).

We agree with the reviewer that our manuscript was very long. To address the reviewer’s concern, we have revised and shortened the manuscript substantially without affecting the quality of our study. The length of the results was cut by approximately 17% (orig. 9 pages and now 7 ½) and the discussion by 40% (orig. 9 pages and now 5 pages).

We appreciate your comments and acknowledgement of the difficulty in obtaining samples. We also wish we could have obtained more samples, especially for the PTMs that had a large variation in values. These variations could result from several causes and more samples may have allowed us to realize additional significant results.

b) The authors also make several statements about how MYH6 or α myosin is induced in human heart failure. To the best of my knowledge, the opposite is what happens in human heart failure.

We thank the reviewer for pointing it out. We will concur that this is a misstatement. We regret the mistake, and the above statement has been removed from the text in the discussion.

2) Reviewer 3:a) I found it very difficult to understand the way in which PTM abundance was presented. It was not clear what the overall percentage of PTM is for a given site, e.g. in a failing heart, what percentage of myosin K951 residues are acetylated? The first paragraph of the discussion states that "phosphorylation and acetylation PTMs are present on β-MHC in non-diseased human hearts in low abundance." Is it not possible to determine what fraction of a particular peptide is modified? Otherwise, why would the authors state that PTMs are "low abundance" (first paragraph of the discussion)? Furthermore, if this information has been obtained, why would it not be clearly represented in the manuscript? It seems as though the quantification of PTMs is shown only relative to IRP.

The PTM abundance was reported as normalized to IRP (internal reference peptide) and not as the percentage of occupancy of a given site. Because the ionization efficiencies of acetylated/phosphorylated peptides are different from the unmodified peptides, we ended up not using the AUC (area under curve) data to calculate the percentage of occupancy. We could have done quantifications in this study but based upon the differences in ionization efficiencies these numerical determinations would not be highly accurate. Instead, we normalized the peptides with PTM to IRP so that we can compare the relative abundance (occurrence) of PTM across all samples. We modified the manuscript to further clarify that we reported the data in this manner. We found one confusing statement (on page 26 of the submitted manuscript) that referred to % occupancy and removed it because we are not able to accurately make these determinations.

Please find Author response table 1 of area under the curve calculations below that show how the normalization of modified to IRP were made. There were no negative values generated.

**Author response table 1. sa2table1:** 

K34	WT	ISC HF	NON-ISC HF	IRP	WT	ISC HF	NON-ISC HF				
	171243328	37933984	123456288		37951213568	35468374016	36720361472		WT	ISC HF	NON-ISC HF
	304486400	474435776	254883488		30835193856	24838664192	28284686336		4.5121964	10.486356	3.36206625
	137056288	474435776	254883488		33189758776	22171635712	32563331072		9.8746387	19.100696	9.011359892
	86855160	136785104	100508704		30544549888	38021373952	37236039680		4.1294752	3.2725629	1.374162978
									2.8435567	3.5975845	2.699231843
								AVG	5.33997	9.1143	4.111705241
K58	WT	ISC HF	NON-ISC HF	IRP	WT	ISC HF	NON-ISC HF				
	356371104	95448224	426329504		37951213568	35468374016	36720361472		WT	ISC HF	NON-ISC HF
	414622496	58387644	338617856		30835193856	24838664192	28284686336		9.3902426	2.6910798	11.61016632
	87149368	56710632	8662002		33189758976	22171635712	32563331072		13.446405	2.3506757	0.233004789
	80678968	142589216	71101432		30544549888	38021373952	37236039680		2.6257909	2.5582069	0.266004789
									2.641354	3.7502384	1.909478898
								AVG	7.02595	2.83755	6.439355856
S210	WT	ISC HF	NON-ISC HF	IRP	WT	ISC HF	NON-ISC HF				
	37987736	30292644	32893104		37951213568	35468374016	36720361472		WT	ISC HF	NON-ISC HF
	149246768	29204212	38378992		30835193856	24838664192	29284686336		1.0009624	0.8540748	0.895772881
	35820280	27507396	54357632		33189758976	22171635712	32563331072		4.8401437	1.1757561	1.356882362
	25008872	18092996	45057196		30544549888	38021373952	37236039680		1.079257	1.240657	1.669289664
									0.8187671	0.4758638	1.210004265
								AVG	1.93478	0.93659	1.282006889
Y215	WT	ISC HF	NON-ISC HF	IRP	WT	ISC HF	NON-ISC HF				
	58306360	60397624	63556984		37951213568	35468374016	36720361472		WT	ISC HF	NON-ISC HF
	214473888	43148672	60914896		30835193856	24838664192	28284686336		1.5363503	1.7028586	1.730837646
		40785388	87614824		33189758976	22171635712	32563331072		6.9554902	1.7371575	2.153635196
	1100631.5	50358960	73574128		30544549888	38021373952	37226039680			1.83953	2.69059771
									0.0360336	1.3244908	1.975884886
								AVG	2.84262	1.65101	2.137738859
K429	WT	ISC HF	NON-ISC HF	IRP	WT	ISC HF	NON-ISC HF				
	38319744	66542720	34278128		37951213568	35468374016	36720361472		WT	ISC HF	NON-ISC HF
	39962528	31968566	25058588		30835193856	24828664192	28284686336		1.0097106	1.8761142	0.933491029
	30577718	29568352	33888364		33189758976	22171635712	32563331072		1.2960038	1.2870485	0.885941873
	26531748	32985240	30807246		30544549888	38021373952	37236039680		0.9212998	1.3336117	1.040690952
									0.8686246	0.8675447	0.82735023
								AVG	1.02391	1.34108	0.921868521
K951	WT	ISC HF	NON-ISC HF	IRP	WT	ISC HF	NON-ISC HF				
	554000765	156454688	274431584		37951213568	35468374016	36720361472		WT	ISC HF	NON-ISC HF
	431275680	104078096	254465008		30835193856	24838664192	28284686336		14.597709	4.411104	7.473555345
	594530432	143491840	150870720		33189758976	22171635712	32563331072		13.986475	4.1901648	8.996564607
	334659520	210390240	44803980		3.05446E+11	38021373952	37236039680		17.913069	6.4718653	4.633147624
									10.95644	5.5334728	1.203242353
								AVG	14.3634	5.15165	5.576627007
K1195	WT	ISC HF	NON-ISC HF	IRP	WT	ISC HF	NON-ISC HF				
	77233384	30782836	20483700		37951213568	35468374016	36720361472		WT	ISC HF	NON-ISC HF
	90984128	70924552	59125832		30835193856	24838664192	28284686336		2.0350702	0.8678953	0.557829476
	44304144	16460342	29519324		33189758976	22171635712	32563331072		2.9506585	2.8554093	2.09038316
	64847704	11629701	423826652		30544569888	38021373952	37236039680		1.3348739	0.7424054	0.906520403
									2.1230532	0.3058727	1.138215889
								AVG	2.11091	1.1929	1.173237232

b) Similarly, the PTM to IRP ratio is also confusing in that the stated range is 1-14. Does this mean that it is not possible for PTM abundance to be less than the IRP? Why not?

Yes, the PTM abundance could be less than the IRP. We performed calculations of PTM relative quantification based on its ratio to IRP and the reported range of 1-14 is what we have observed. The ratio depends on the abundance of PTM relative to IRP.

c). The authors mention that one way to interpret the loss of K951 acetylation in failing hearts is that there is a shift toward α-MHC. Perhaps it was outside the scope of the study, but was there no way of testing this hypothesis in some way?

While this statement is accurate in context with the shifts that occur in mouse hearts which has a different dominant isoform, a-MHC (mouse) and b-MHC (human) the upregulation of a-MHC in human heart failure does not occur as stated. Therefore, we removed these several statements from the text regarding an isoform switch. A more likely scenario underlying the reduction of K951-Ac in failing hearts is upregulation of HDACs in ischemic and non-ischemic heart failure, which has been documented by a number of groups.

d). The discussion is, in my view, far too long and wide-ranging. Some of the logic presented is also challenging. For instance, the PTMs discussed are shown to be more abundant in normal than in failing hearts. Yet, the authors draw parallels between the sites for the PTMs and the locations of cardiomyopathy-causing mutations. It seems strange to associate sites of "normal" PTMs with the locations of disease-causing mutations. Overall, the discussion is far longer than would normally be expected for the corresponding significant results, which really focused just on K951. My recommendation would be to focus the discussion drastically, state the most meaningful findings very clearly, and help the reader grasp the real significance of the work.

We agree with the reviewer that our manuscript was very long. To address the reviewer’s concern, we have revised and shortened the manuscript substantially without affecting the quality of our study. The length of the results was cut by approximately 17% (orig. 9 pages and now 7 ½) and the discussion by 40% (orig. 9 pages and now 5 pages).

Our inclusion of a Table of CM mutation was intended to demonstrate importance of altering interactions/function of these regions due to inclusion of a PTM at that site. It is intended to be merely correlative; we understand your concern about including this data for “normal” PTMs, but in our view the PTMs may represent a “gain-of-function” or promote enhanced contractility –

We shortened but also rearranged the discussion so that it is now organized in a way that emphasized our major findings and highlighting the real significance of the work.

Reviewer #2 (Recommendations for the authors):This paper tackles an important understudied problem of PTMs in the human β cardiac myosin protein. The authors found acetylation sites and phosphorylation sites and used molecular dynamics simulations to predict the impacts of these PTMs on myosin function.There is a lot of information in this very long (too long in my opinion) right now, its utility is unknown. The PTMs are very rare on this molecule and there are not enough samples to have much power (even though I know how hard it is to get samples).The authors also make several statements about how MYH6 or α myosin is induced in human heart failure. To the best of my knowledge, the opposite is what happens in human heart failure.I think this information may be better suited for a more specialized audience and journal.

We appreciate your critique and the opportunity to improve our manuscript and make it more accessible to the general reader. We have thoughtfully and carefully considered each point that you have made. We believe that the manuscript is now greatly improved and suitable for publication in *eLife*.